# Methane-hydrogen-rich fluid migration may trigger seismic failure in subduction zones at forearc depths

Francesco Giuntoli [1] ✉, Luca Menegon[2], Guillaume Siron [1],
Flavio Cognigni [3], Hugues Leroux[4], Roberto Compagnoni[5], Marco Rossi [3] &
Alberto Vitale Brovarone[1,6,7] ✉

Metamorphic fluids, faults, and shear zones are carriers of carbon from the deep Earth to shallower reservoirs. Some of these fluids are reduced and transport energy sources, like $H_2$ and light hydrocarbons. Mechanisms and pathways capable of transporting these deep energy sources towards shallower reservoirs remain unidentified. Here we present geological evidence of failure of mechanically strong rocks due to the accumulation of $CH_4$-$H_2$-rich fluids at deep forearc depths, which ultimately reached supralithostatic pore fluid pressure. These fluids originated from adjacent reduction of carbonates by $H_2$-rich fluids during serpentinization at eclogite-to-blueschist-facies conditions. Thermodynamic modeling predicts that the production and accumulation of $CH_4$-$H_2$-rich aqueous fluids can produce fluid overpressure more easily than carbon-poor and $CO_2$-rich aqueous fluids. This study provides evidence for the migration of deep Earth energy sources along tectonic discontinuities, and suggests causal relationships with brittle failure of hard rock types that may trigger seismic activity at forearc depths.

Subduction zones are major geological structures where life-essential elements, such as H, O, and C, are mobilized by metamorphic fluids[1]. In forearc regions, which also represent the most hazardous part of convergent margins, tectonic discontinuities promote the eventual return of these fluids to the Earth's surface and the preservation of habitable conditions in the sub-surface biosphere[2–4]. In turn, fluid migration may promote seismic activity along subduction zones, for example, causing pore pressure fluctuations[5,6].

Carbon-bearing fluids are widespread in subduction zones and their speciation is sensitive to the redox state of geological reservoirs[7,8]. Under oxidized conditions, $CO_2$ and/or other oxidized C-bearing species may be present, while under reduced conditions

$CH_4$ and $H_2$ are expected to be dominant[9]. It is generally accepted that the presence of aqueous fluids weakens rocks, due to the presence of free aqueous fluid at grain boundaries that can facilitate hydrolytic- and reaction weakening during viscous deformation[10], or hydrofracturing[5,11]. Likewise, the infiltration of carbon-bearing aqueous fluids may affect the rheology of the country rock, although its effects are more debated. Experimental studies show that $CO_2$-rich aqueous fluids produce (i) negligible effects on strength when interacting with basaltic rocks at conditions compatible with $CO_2$ storage in basaltic reservoirs[12], (ii) strengthening or weakening of quartzite, depending on the oxygen fugacity, in high-pressure and temperature experiments[13] and (iii) fracturing in quartzite and dunite, as shown by

[1]Department of Biological, Geological, and Environmental Sciences, Università degli Studi di Bologna, Bologna, Italy. [2]The Njord Centre, Department of Geosciences, University of Oslo, Oslo, Norway. [3]Department of Basic and Applied Sciences for Engineering (SBAI), Università degli Studi di Roma La Sapienza, Rome, Italy. [4]Univ. Lille, CNRS, INRAE, Centrale Lille, UMR 8207, UMET, Unité Matériaux et Transformations, Lille, France. [5]Dipartimento di Scienze della Terra, Università degli Studi di Torino, Torino, Italy. [6]Sorbonne Université, Muséum National d'Histoire Naturelle, UMR CNRS 7590, IRD, Institut de Minéralogie, de Physique des Matériaux et de Cosmochimie, IMPMC, Paris, France. [7]Institute of Geosciences and Earth Resources, National Research Council of Italy, Pisa, Italy. ✉e-mail: francesco.giuntoli@unibo.it; alberto.vitaleb@unibo.it

high-temperature experiments[14]. Studies of natural anorthosite in lower crustal shear zones evidence that $CO_2$-rich conditions promote metamorphic reactions and precipitate fine-grained metamorphic minerals at grain boundaries, thus limiting the average grain size and favoring grain size-sensitive creep[15]. In extensional settings, $CO_2$-rich fluid migration may produce metasomatism and strain softening in peridotite, also leading to seismic activity[16]. In subduction zones, $CO_2$-rich fluids can produce metasomatism of the hydrated mantle and trigger fracturing by dehydration reactions and volume changes[17]. $CO_2$-rich fluid can accumulate under low-permeability rocks leading to high pore pressure and seismic activity at crustal depth[18–23]. Additionally, the larger wetting angles of $CO_2$ compared to an aqueous fluid (~90° for the former and ~57–67° for the latter, depending on the mineral assemblage[24]) together with fluid immiscibility can segregate $CO_2$ from the aqueous fluid and generate preferential $CO_2$ migration via hydrofracturing[25–29]. Finally, $CO_2$-rich fluids expand more than aqueous fluids during the exhumation and decompression of subducted rocks, leading to failure in low-permeability rock types[30].

The effect of the infiltration of more reduced $CH_4$-$H_2$-rich aqueous fluids on the stability and rheology of rocks is, in contrast, less studied at crustal and mantle depths, even though reducing conditions can be locally dominant in subduction zones[31–34]. Compared to $CO_2$, $CH_4$, and $H_2$ are energy sources playing a fundamental role in metabolic processes in the subsurface biosphere. The mechanisms capable of promoting the migration of these fluids towards shallower reservoirs, including the biosphere, remain largely unknown. Nevertheless, it has recently been documented that the genesis of $CH_4$-rich fluids causes strain localization and may promote seismic activity in subducted rocks[35], processes through which the migration of these energetic fluids may be favored.

In this contribution, we show evidence of brecciation of omphacitite, a rock rich in omphacite that represents the characteristic clinopyroxene of eclogite-facies mafic rocks. This rheologically strong rock type is in contact with serpentinites and carbonated serpentinites from which $CH_4$-$H_2$-rich fluids were produced at eclogite-facies conditions[31,35,36]. Mineral assemblages constrain brecciation to a depth range of 30–80 km in the Alpine subduction zone, thus consistent with the $CH_4$ formation conditions. We suggest that the migration and accumulation of $CH_4$-$H_2$-rich fluids below a permeability seal, possibly promoted by $CH_4$ and $H_2$ immiscibility and segregation in aqueous fluids, led to supralithostatic fluid pressure conditions[37] that triggered brittle, and potentially seismic, failure of the host rock, which further facilitated the migration of these fluids in the subduction zone.

## Results
### Geological and structural setting
The Lanzo Massif is a lithospheric mantle body of ~150 km² situated in the Western Italian Alps. The massif consists of a core of fresh peridotite rimmed by serpentinites that experienced a complex evolution, from Jurassic exhumation and alteration at the seafloor, to Eocene subduction and Alpine high-pressure metamorphism at ~550–600 °C and 2–2.5 GPa[38] (Fig. 1a). The serpentinized rim include bodies of carbonated serpentinites (also called ophicarbonates), which are interpreted to have formed prior to Alpine subduction, during the hydrothermal alteration at the seafloor[38,39]. Nevertheless, another alteration event occurred at high-pressure conditions in the Alpine subduction zone and resulted in the serpentinization of a fresh peridotite portion and the associated genesis of $H_2$-rich fluids[31]. This alteration is evidenced by the fluid-mediated reduction of a carbonated serpentinite body, which resulted in the formation of abiotic $CH_4$ and graphite at near peak metamorphic conditions, around $49.6 \pm 1.0$ Ma, and during exhumation[31,36]. This process was interpreted to reflect the availability of $H_2$-rich reduced fluids formed through serpentinization and their percolation inside the carbonated

serpentinite at high-pressure conditions[34]. The process of carbonate reduction and $CH_4$ formation was shown to cause strain localization inside the reacted carbonated serpentinite[35]. The presence of graphite-rich veins inside the surrounding serpentinite was used to suggest that the $CH_4$ produced inside the carbonated serpentinite was capable of migrating outwards and precipitating graphite[31].

In this article, we investigate the mechanical behavior of rocks adjacent to the reacted carbonated serpentinite in response to the migration and accumulation of these methane-rich metamorphic fluids. In the study area, the serpentinized Lanzo Massif is separated from the adjacent continental Sesia Zone by an omphacite-rich layer, hereafter omphacitite[40–42] (Fig. 1). The origin of this rock type, which is mineralogically similar to an eclogite, is ascribed to metasomatic processes affecting the interface between the Lanzo serpentinite and the Sesia gneissic rocks[36–38]. This work focuses on the mechanical behavior of this rock, as a case study similar to the eclogitized mafic crust forming atop of subducting lithospheric mantle sections, whereas its metasomatic origin is beyond the scope of this work and will not be discussed further. The omphacitite layer can be observed with a lateral continuity of a few hundred meters and a thickness up to several meters. Tectonic slicing of this contact produced a few isolated layers of omphacitite within serpentinites (Fig. 1b)[30,34]. The studied omphacitite lens is up to 5 meters thick and tens of meters long and is bounded by serpentinites. It is located at <20 meters (across the main foliation) from the reduced carbonated serpentinite from which intense eclogite-facies metamorphic $CH_4$ production was previously identified[31,35]. In this area, the omphacitite shows intense brecciation evidenced by omphacitite angular clasts and fractures sealed by a dark filling material, over a thickness of >2 meters and a length of >5 meters (Fig. 1c, e). Discontinuous talcschist layers occur both at the contact between omphacitite and serpentinite and within the omphacitite (Fig. 1d). Talcschist layers have a maximum thickness of 30 cm and a length of several meters. The main foliation inside the serpentinite, omphacitite, and talcschist is parallel to the main lithological boundaries and dips with low angles to the E-NE.

### Microstructural and microchemical constraints of dilation breccia formation
X-Ray Microscopy and petrographic analysis highlight a dilation breccia structure, with internally weakly foliated omphacitite fragments ranging in size from a few microns to several centimetres (Figs. 2 and 3, Supplementary Figs. 1–4, Supplementary information X-ray Microscopy (XRM) statistical data and Supplementary Movies 1-6). These fragments are sealed by a pervasive and interconnected matrix composed of variable proportions of jadeite, omphacite, grossular, titanite, and graphite (Fig. 4c–h and Supplementary Figs. 5–8). The omphacitite breccia can be classified as crackle and mosaic breccia[43]. Locally, the breccia shows also jadeitite veins oriented subparallel to the foliation visible in the clasts. The veins are made of euhedral jadeite grains growing syntaxially, suggesting a crack-and-seal growth[44] (green arrows in Supplementary Fig. 1). Post-brecciation ductile deformation is represented by the local eclogite-facies foliation reworking both the clasts and the matrix. Near the contact with the serpentinite, the omphacitite breccia is locally overprinted by a mylonitic foliation rich in chlorite ± pumpellyite developed at retrograde, greenschist-facies conditions (Supplementary Fig. 4e, f).

Graphite, as identified by Raman spectroscopy (Supplementary Fig. 5), forms continuous or discontinuous rims around the clasts, with a maximum thickness of 50 μm. Graphite is also located in the matrix, along discrete fractures cutting the clasts and along more diffuse fractured areas located at the edges of the clasts (Figs. 2a, 4a,b). Interpenetrated clasts show graphite-enriched stylolitic structures mainly parallel to the post-brecciation eclogite-facies foliation (yellow arrows in Figs. 1e, 4a,b and Supplementary Figs. 1,2,3). Three generations of omphacite are distinguished based on microstructural

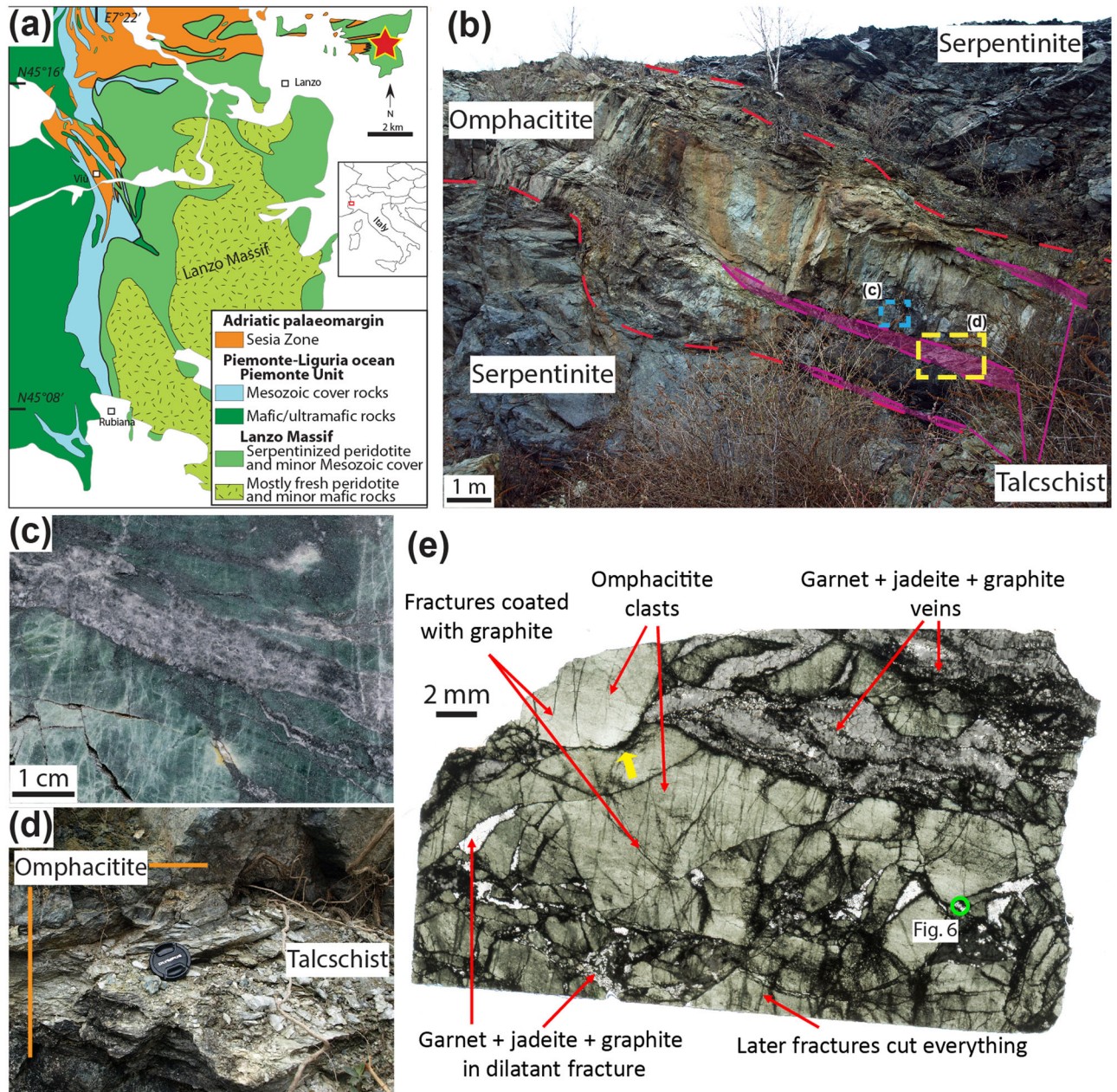

Fig. 1 | Geological map, field images and microstructures of the studied omphacitite. a Geological map of the Lanzo Massif, Italian Western Alps. The star highlights the location of the outcrop (GPS coordinates: 45°17'44"N 7°30'36"E), the red square the approximate geological map location (modified from[31]). Figure created with Adobe Illustrator CS6 (https://www.adobe.com/products/illustrator.html). b Outcrop of omphacitite layers, a few meters thick and tens of meters long, bounded by serpentinites and discontinuous talcschist-rich layers. c Polished slab of omphacitite displaying a fracture network (light green) and graphite-garnet-bearing jadeitite veins (silvery gray). d Enlargement of the contact between omphacitite and talcschist-rich layer. e Thick section scan highlighting the brec-ciated structure, with omphacitite clasts surrounded by a matrix of jadeite, omphacite, grossular, titanite, and graphite. The yellow arrow shows a graphite-enriched stylolitic structure between two interpenetrated clasts. The location of fluid inclusion shown in Fig. 6a.

observations and electron probe micro-analyser chemical data. Gen-eration 1 and 2 are located inside the clasts, and represent fractured omphacite cores and their overgrowth, respectively (FeO >6.4 wt%, $X_{Jd}$ 0.4 $X_{Acm}$ 0.1 $X_{Aug}$ 0.5 and FeO 4.8- 6.4 wt%, $X_{Jd}$ 0.41 $X_{Acm}$ 0.09 $X_{Aug}$ 0.5, respectively; Table 1 and Fig. 2b,d and Supplementary Figs. 9-12). The second generation is more abundant toward the clast edges. Frag-ments of the first generation dispersed in the breccia filling matrix display a rim of the second generation in optical continuity, suggesting epitaxial overgrowth (arrow in Fig. 2d and S8f,g). The third generation is located either along discrete fractures cutting the clasts, or in the matrix. In the latter case, omphacite-3 forms dendritic/acicular inter-grown with jadeite and displays the lowest FeO values (2-6 wt%, $X_{Jd}$

0.42 $X_{Acm}$ 0.12 $X_{Aug}$ 0.45; Figs. 2d and 4c-f, Supplementary Fig. 6). Jadeite in the matrix displays rather pure compositions both for idio-blastic grains and for dendritic grains intergrown with the third gen-eration of omphacite ($X_{Jd}$ 0.91 $X_{Acm}$ 0.08 $X_{Aug}$ 0 and $X_{Jd}$ 0.95 $X_{Acm}$ 0 $X_{Aug}$ 0.05, respectively; Table 1 and Supplementary Figs. 7 and 9c,d). Grossular garnet in the matrix displays complex cauliflower-like habi-tus intergrown with jadeite and complex oscillatory zoning, with some grains showing increasing grossular and decreasing andradite content toward the rims ($X_{Grs}$ 0.93 $X_{Adr}$ 0.06, and $X_{Grs}$ 0.95 $X_{Adr}$ 0.04, respectively), whereas other grains displaying opposite zoning (Fig. 4g,h and Supplementary Figs. 9,11,13). The presence of jadeite and omphacite crystallizing in the matrix and the absence of albite

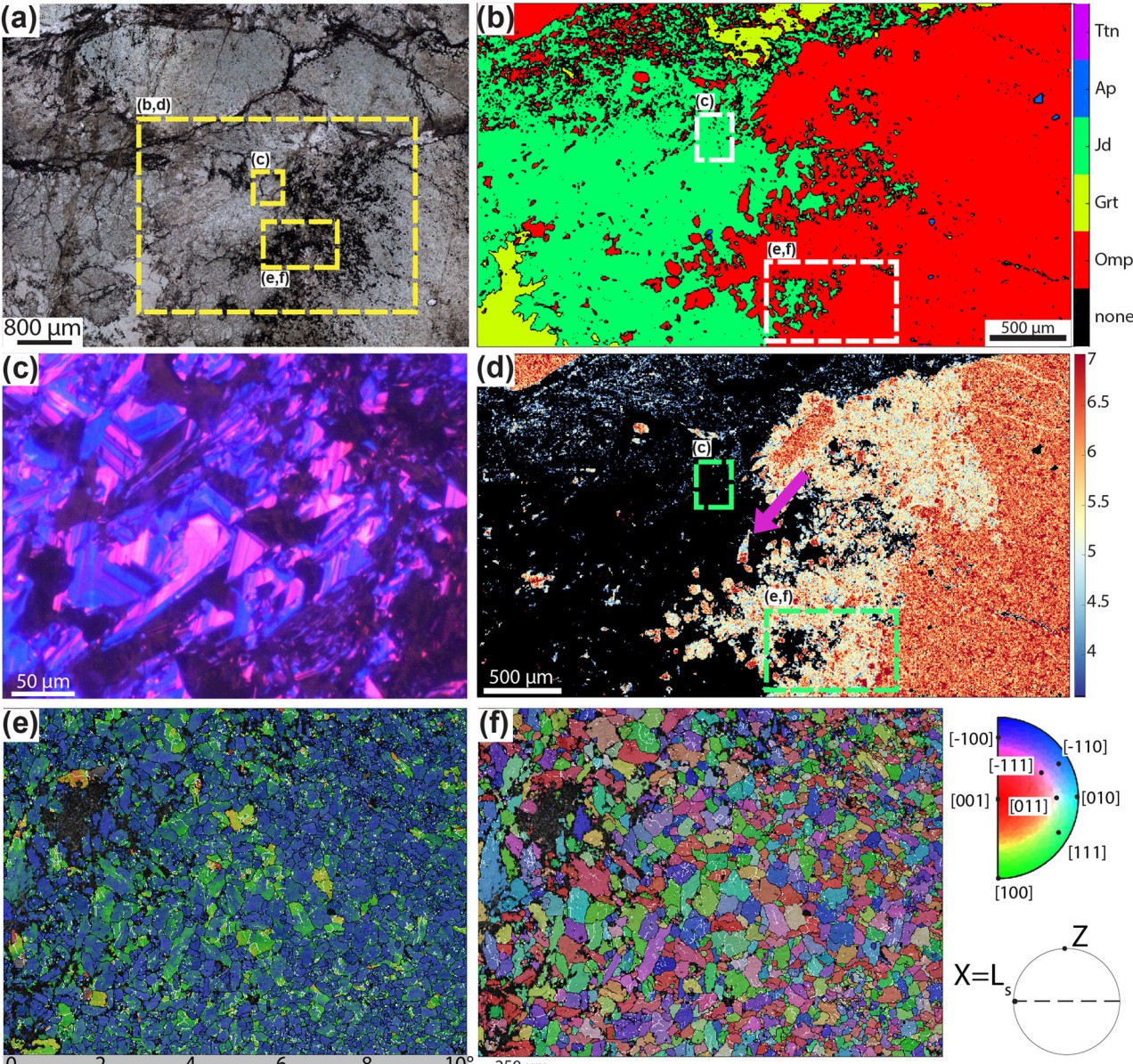

**Fig. 2 | Microstructures, cathodoluminescence, chemical and electron back-scatter diffraction maps of the studied omphacitite. a** Thin section image showing the contact between omphacitite clasts and matrix. Graphite rims omphacitite clasts and is located inside the matrix and along discrete fractures cutting the clasts (Plane-polarized light). See location in Supplementary Fig. 1. **b** X-ray map color coded for the different mineral phases. Ttn: titanite, Ap: apatite, Jd: jadeite, Grt: garnet, Omp: Omphacite. **c** CL image highlighting oscillatory growth zoning in jadeite grains located in the matrix. **d** Standardized X-ray map of the FeO weight % in omphacite. Note the decrease of the FeO content toward the rim of the clast. **e**, **f** EBSD maps. In all the EBSD maps white lines represent low-angle boundaries (2–10° of misorientation), black lines represent high angle boundaries (misorientation > 10°), X is parallel to the stretching lineation (Ls) and Z parallel to the pole of the foliation. **e** GROD map highlighting irregular grain boundaries and low (<5°) internal lattice distortion. **f** IPF map, color-coded with respect to the X-kinematic axis, displays no crystallographic preferred orientation.

suggests at least 1 GPa and 400 °C for the fluid-mediated brecciation[45]. Noteworthy, the methane formation in the massif was constrained between 2 GPa and 550 °C and 1 GPa and 400 °C[31,36].

Scanning electron microscope (SEM) electron backscatter diffraction (EBSD) analysis was performed close to the edge of an omphacitite clast to determine if a stage of crystal-plastic loading preceded brecciation and to constrain the grain size distribution of the brecciated domains (Fig. 2e, f and Supplementary Fig. 14). Omphacite grain size (measured as the diameter of the equivalent circle) ranges between a few microns to a maximum of 100 microns, with an average value of 19 microns. An exponential decrease is visible in the plot of the frequency versus the grain size of grains (Supplementary Fig. 14e).

Larger omphacite grains display irregular grain boundaries and abundant low angle boundaries. Grain reference orientation deviation (GROD) map displays generally low (< 5°) internal lattice distortion, with most of the grains having values < 2° (Fig. 2e). Inverse pole figures (IPF) map and pole figures highlight that grains do not show a crystallographic preferred orientation (Fig. 2f). The misorientation angle distribution (MAD) of correlated pairs shows major peaks at low angle misorientations (between 2° and 10°) and minor peaks for misorientations values < 30° compared to the theoretical random distribution. A few isolated peaks are present for uncorrelated pairs for misorientations comprised between 60° and 160° (Supplementary Fig. 14f). The peaks located at low angle misorientations are

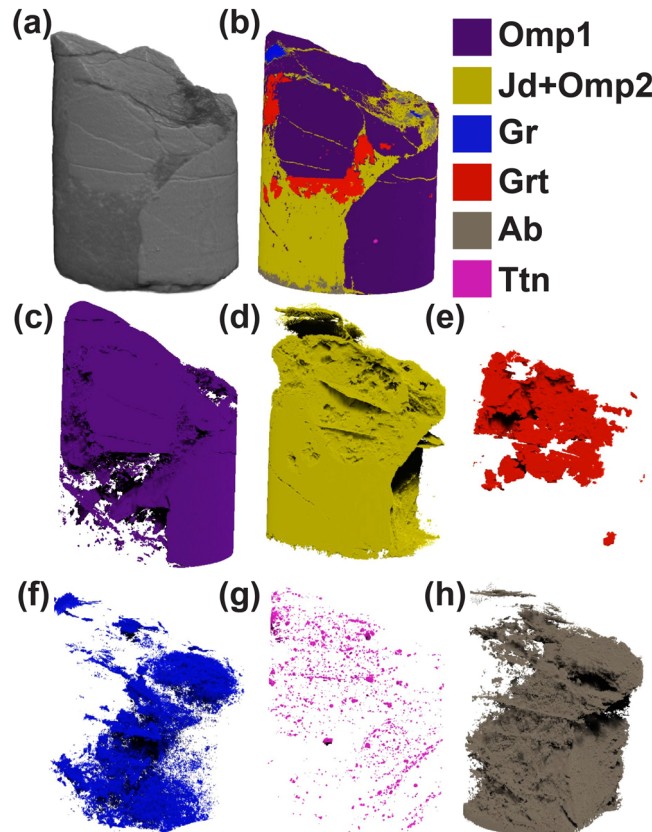

**Fig. 3 | X-ray Computed Tomography of omphacitite core (size of sample: 5.2 mm height, 3.9 mm diametre). a, b** XRM highlights a pervasive network of interconnected fractures. The fractures are sealed by jadeite, garnet, omphacite 2 and graphite. Albite partially replaces jadeite during exhumation. **c–h** Display of single mineral phases color-coded as in legend. Mineral abbreviations as in the previous figure caption, with additional Ab: albite, Gr: Graphite.

interpreted to reflect fracturing of former single grains with negligible offset and grain rotation[46]. These data indicate diffuse microfracturing of omphacite grains with minimal role of crystal plasticity prior to and during the brecciation event.

Some omphacite grains located close to the edges of the clasts are twinned, whereas no twinned grains were observed in the matrix (Fig. 5 and Supplementary Fig. 11i, j). Transmission Electron Microscope (TEM) analysis shows that these omphacite grains are highly twinned on (100) planes (Fig. 5c–e). The thickness of the twins is highly variable, from a few tens to a few hundreds of nm. The twin walls contain regularly spaced dislocations testifying to the mechanical origin of these twins, which is typically observed to form at critical resolved shear stress higher than 100 MPa[47–49] (see below for discussion). Some inclusions of carbon-rich amorphous material are located along the twins (Fig. 5e). Their size is around a hundred nm and their shape is elongated in the direction of the twins. Reduced carbon precipitation from the fluid postdates the twinning, as the twinning planes may act as a local preferential weakness where fluid could migrate[50]. Additionally, the presence of $H_2$-rich fluids may have enhanced the reworking of such discontinuities, as documented in materials science[51]. Finally, this chronology is also supported by the presence of carbon in the breccia matrix that is, again, successive to twinning.

SEM Back-Scattered Electron (BSE) imaging highlights dendritic textures of the garnet, jadeite and omphacite intergrows sealing the breccia fragments (Fig. 4). These growth structures in metamorphic rocks suggest fast precipitation under disequilibrium conditions[52–55]. Cathodoluminescence (CL) imaging of euhedral jadeite and garnet in the matrix shows oscillatory zoning (Fig. 2c and Supplementary

Figs. 8,10,11,13), as observed in jadeitites by Harlow et al.[45], and suggests precipitation from multiple fluid pulses[56] or from an evolving fluid not in equilibrium with the surrounding rock.

Raman Spectroscopy highlights $CH_4$, $H_2$, and $H_2O$ fluid inclusions located in the euhedral jadeite sealing the breccia (Fig. 6). Based on the collected Raman spectra, the gas bubble in the fluid inclusions consists of 86 mol% of $CH_4$ and 14 mol% $H_2$. No $CO_2$ was detected. Using the obtained $X_{CH_4}$ and $H_2$ proportions, we calculated the $X_{CH_4}$ ($CH_4$/($CH_4 + CO_2 + H_2O + H_2$)) of the trapped fluid. The calculation did not consider fluid-inclusion respeciation during exhumation. The results suggest that the fluid trapped in the fluid inclusions had an $X_{CH_4} = 0.65$ ($XH_2O = 0.25$; $XH_2 = 0.1$; Supplementary Fig. 15). This corresponds to strongly reducing conditions – $fO_2$ 7 units below the fayalite-magnetite-quartz (FMQ) buffer. This $fO_2$ value is similar to previous estimates on the $CH_4$-forming conditions in the adjacent carbonated serpentinite[31], and indicate that the fluid present during the brecciation event was a $CH_4$-rich aqueous fluid (see also discussion below). This feature suggests a link between $CH_4$ and $H_2$-rich fluids originated in the adjacent carbonated serpentinite at eclogite-facies conditions and those present during omphacite brecciation.

## Discussion

The eclogite-like, omphacitite metasomatic horizon bounding the Lanzo serpentinites displays evidence of ductile and brittle deformation occurred at near peak, eclogite-facies, to early retrograde, blueschist-facies conditions (2 GPa and 550 °C and 1 GPa and 400 °C) during Alpine subduction. The formation of dilation breccia attests for high pore pressure[57]. Fluid inclusions and the presence of fluid-deposited graphitic carbon in the breccia matrix indicate that the fluid was aqueous and $CH_4$ and $H_2$-rich. The matrix microstructures, including dendritic and cauliflower-like crystal growths, suggest fast sealing in plausible disequilibrium conditions[53–55]. The relationships between $CH_4$-bearing fluid circulation and brecciation can be derived from the relative chronology of deformation and metamorphic events reconstructed in the natural samples, which here below are discussed starting from the oldest.

Mechanical twinning of clinopyroxene grains located in the omphacitite clasts pre-dated the brecciation event and the ingression of carbon-bearing fluids. Twinning of clinopyroxene requires high differential stresses, with a critical resolved shear stress around 140–150 MPa, as reported in refs. 47,58. Assuming 140 MPa as critical shear stress, Trepmann & Stockhert[48] found differential stresses ($\Delta\sigma$) of 0.5 GPa for twinned jadeite grains in metagranite from Montestrutto (Western Italian Alps), with comparable results also obtained for omphacite[49]. These authors interpreted the high $\Delta\sigma$ as peak stresses resulting from synseismic loading. In the studied omphacitite, Mohr-Coulomb diagrams show that brittle failure in a thrust regime would occur at $\Delta\sigma$ of -1.4 and 3.4 GPa for low pore fluid factor approaching hydrostatic values ($\lambda = 0.4$) and vertical stresses ($\sigma_V$) of 1 and 2 GPa, respectively, corresponding to the minimum compressive stress ($\sigma_3$) during deformation and metamorphism of the omphacitite (see section "Microstructural and microchemical constraints of dilation breccia"; Fig. 7). $\Delta\sigma$ of more than 1 GPa were most likely not attained, but instead $\Delta\sigma$ in the order of some hundreds of MPa triggered mechanical twinning in omphacite. The mechanical twinning is interpreted to represent a prerupture loading event, thus a strain response to instantaneous stress changes[48,58].

The infiltration of carbon-bearing reduced fluids and the brecciation rapidly post-dated the twinning, as carbon precipitated along the twin boundaries of the strongly deformed omphacite crystals and in the breccia matrix (Fig. 5). The successive dilation brecciation of omphacitite could require much lower $\Delta\sigma$ values compared to the case of fracturing under hydrostatic $\lambda$ values, provided that supralithostatic $\lambda$ values were reached. For $\sigma_V$ of 1 and 2 GPa, shear failure occurs at $\Delta\sigma$ of 300 MPa for sublithostatic $\lambda$ values of 0.9–0.97, respectively, hybrid

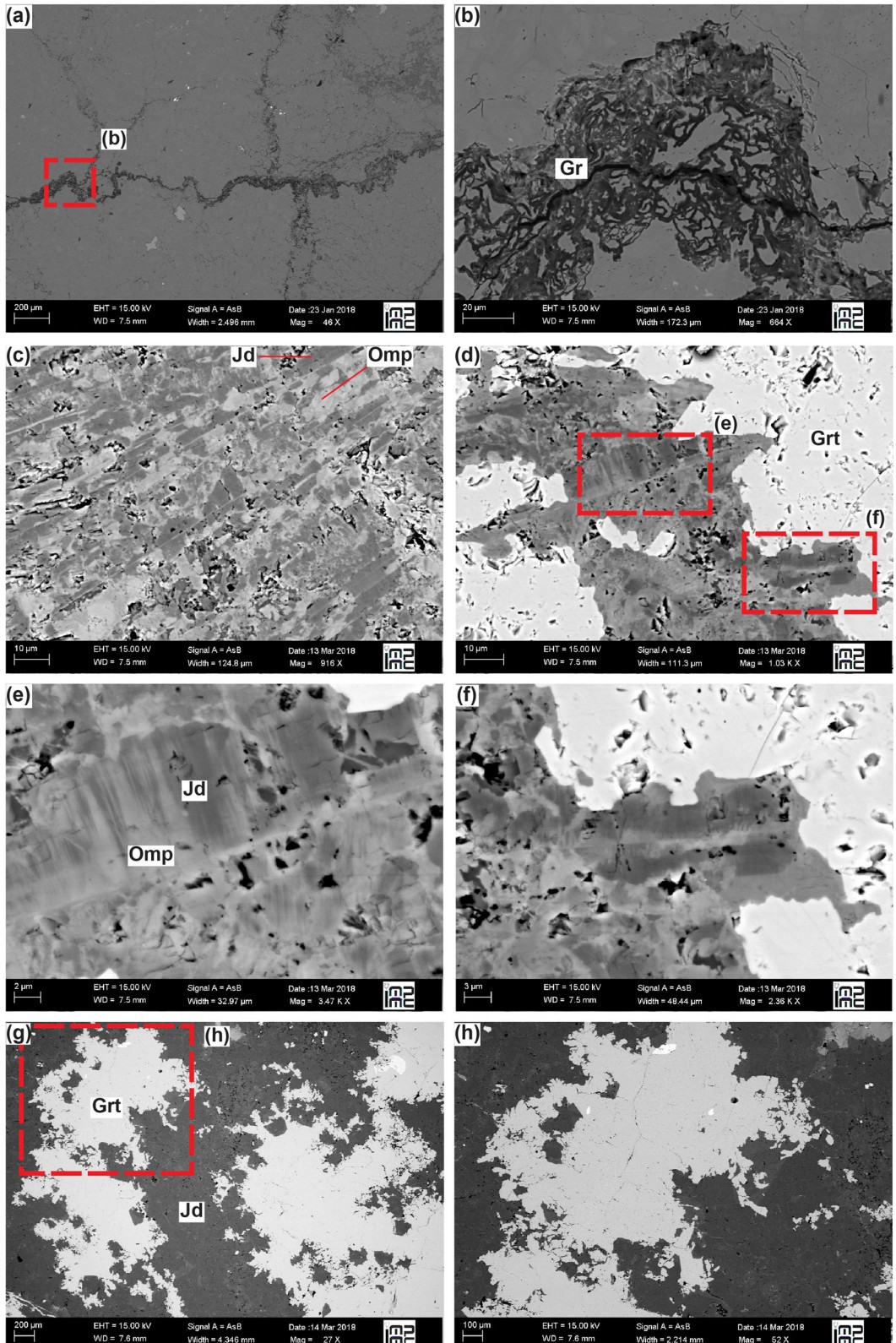

**Fig. 4 | Backscattered-electron images of omphacitite. a, b** Graphite-enriched stylolitic structures within and between omphacitite clasts and **c–h** dendritic/acicular textures of omphacite, jadeite and cauliflower garnet intergrown in the matrix indicating fast growth and disequilibrium textures.

failure occurs at Δσ of 114 MPa for supralithostatic λ values of 1.02–1.01 and extension failure occurs at Δσ of 100 MPa for same λ values (Fig. 7). High pore pressure is also supported by the fact that omphacite twinning is rare and not pervasive, differently to what observed in dry lower crustal conditions, where clinopyroxene deformation twinning

is pervasive along pseudotachylite-bearing faults formed at high differential stress and low pore pressure[59]. Accordingly, the formation of the omphacitite breccia requires pore pressure increase to supralithostatic values. Fluid inclusions in the breccia matrix, as well as the precipitation of graphitic carbon in it, indicate that the overpressured

**Table 1 | Representative spot and average composition analyses (wt%) of the mineral phases**

Sample: 1BAl17-4a1Nuova

| Wt% | Garnet Spot | Garnet Core Av Comp | Garnet Core σ | Garnet Rim Av Comp | Garnet Rim σ | Omphacite Spot Hi FeO | Omphacite Spot Low FeO | Omphacite Core Av Comp | Omphacite Core σ | Omphacite Rim Av Comp | Omphacite Rim σ | Omphacite Intergrown with Jd Av Comp | Omphacite Intergrown with Jd σ | Jadeite Spot Low FeO | Jadeite Spot Hi FeO | Jadeite Euhedral Av Comp | Jadeite Euhedral σ | Jadeite Intergrown with Omp Av Comp | Jadeite Intergrown with Omp σ | Titanite Spot |
|---|---|---|---|---|---|---|---|---|---|---|---|---|---|---|---|---|---|---|---|---|
| $SiO_2$ | 37.86 | 38.31 | 0.79 | 38.57 | 0.72 | 54.31 | 55.20 | 53.67 | 1.33 | 53.46 | 1.42 | 53.99 | 0.96 | 58.55 | 57.70 | 57.78 | 1.00 | 57.71 | 0.85 | 29.75 |
| $TiO_2$ | 0.90 | 0.86 | 0.08 | 0.18 | 0.03 | 0.01 | 0.09 | 0.03 | 0.01 | 0.03 | 0.01 | 0.03 | 0.01 | 0.01 | 0.05 | 0.02 | 0.00 | 0.03 | 0.01 | 39.23 |
| $Al_2O_3$ | 21.23 | 21.37 | 0.57 | 22.15 | 0.69 | 10.50 | 11.99 | 10.25 | 0.80 | 10.90 | 0.78 | 11.82 | 0.55 | 25.62 | 22.07 | 25.75 | 0.76 | 22.62 | 0.68 | 1.23 |
| FeO | 1.97 | 1.99 | 0.23 | 1.37 | 0.23 | 6.97 | 4.81 | 7.67 | 0.74 | 6.38 | 0.45 | 4.82 | 0.42 | 0.04 | 2.30 | 0.16 | 0.06 | 0.49 | 0.11 | 0.25 |
| MnO | 0.17 | 0.21 | 0.06 | 0.18 | 0.05 | 0.36 | 0.13 | 0.29 | 0.09 | 0.20 | 0.06 | 0.18 | 0.05 | 0.04 | 0.00 | 0.02 | 0.01 | 0.02 | 0.01 | – |
| MgO | 0.01 | 0.02 | 0.01 | 0.02 | 0.00 | 6.57 | 6.97 | 5.53 | 0.38 | 6.05 | 0.40 | 6.59 | 0.35 | 0.04 | 1.32 | 0.12 | 0.05 | 0.46 | 0.04 | 0.01 |
| CaO | 37.96 | 37.98 | 0.92 | 38.39 | 0.68 | 14.03 | 13.16 | 13.62 | 0.92 | 13.39 | 0.79 | 12.84 | 0.63 | 0.15 | 2.44 | 0.29 | 0.13 | 1.21 | 0.14 | 28.90 |
| $Na_2O$ | – | – | – | – | – | 6.99 | 7.79 | 7.00 | 0.49 | 7.06 | 0.50 | 7.80 | 0.33 | 15.64 | 14.22 | 15.67 | 0.35 | 13.87 | 0.31 | – |
| $K_2O$ | – | – | – | – | – | – | – | – | – | – | – | – | – | – | – | – | – | – | – | – |
| Total | 100.10 | 100.75 | 1.37 | 100.86 | 1.23 | 99.74 | 100.14 | 98.07 | 1.85 | 97.47 | 1.97 | 98.06 | 1.42 | 100.09 | 100.10 | 99.81 | 1.31 | 96.39 | 1.15 | 99.37 |
| Formulae based | on 12 O | | | | | on 6 O | | | | | | | | | | | | | | |
| Si | 2.86 | 2.89 | 0.04 | 2.90 | 0.04 | 1.95 | 1.95 | 1.97 | 0.03 | 1.97 | 0.03 | 1.96 | 0.02 | 1.96 | 1.96 | 1.97 | 0.02 | 2.00 | 0.00 | |
| Ti | 0.05 | 0.05 | 0.01 | 0.01 | 0.00 | 0.00 | 0.00 | – | – | – | – | – | – | 0.00 | 0.00 | – | – | – | – | |
| Al | 1.89 | 1.90 | 0.05 | 1.96 | 0.05 | 0.45 | 0.50 | 0.45 | 0.06 | 0.47 | 0.06 | 0.51 | 0.04 | 1.01 | 0.88 | 1.03 | 0.02 | 0.95 | 0.02 | |
| $Fe^{3+}$ | 0.12 | 0.12 | 0.02 | 0.09 | 0.02 | 0.13 | 0.13 | 0.08 | 0.05 | 0.07 | 0.05 | 0.08 | 0.03 | 0.00 | 0.07 | 0.00 | 0.00 | 0.00 | 0.00 | |
| $Fe^{2+}$ | 0.00 | 0.00 | 0.01 | 0.00 | 0.01 | 0.07 | 0.02 | 0.16 | 0.05 | 0.13 | 0.05 | 0.06 | 0.03 | 0.00 | 0.00 | 0.00 | 0.01 | 0.01 | 0.01 | |
| Mn | 0.01 | 0.01 | 0.00 | 0.01 | 0.00 | – | – | – | – | – | – | – | – | – | – | – | – | – | – | |
| Mg | 0.00 | 0.00 | 0.00 | 0.00 | 0.00 | 0.35 | 0.37 | 0.30 | 0.02 | 0.33 | 0.02 | 0.36 | 0.02 | 0.00 | 0.07 | 0.01 | 0.00 | 0.02 | 0.00 | |
| Ca | 3.07 | 3.07 | 0.07 | 3.09 | 0.05 | 0.54 | 0.50 | 0.54 | 0.03 | 0.53 | 0.03 | 0.50 | 0.03 | 0.01 | 0.09 | 0.01 | 0.01 | 0.05 | 0.01 | |
| Na | – | – | – | – | – | 0.49 | 0.53 | 0.50 | 0.03 | 0.50 | 0.03 | 0.55 | 0.03 | 1.02 | 0.94 | 0.98 | 0.01 | 0.93 | 0.02 | |
| K | – | – | – | – | – | – | – | – | – | – | – | – | – | – | – | – | – | – | – | |
| Σ cations | 8.00 | 8.05 | – | 8.07 | – | 3.99 | 4.00 | 4.00 | – | 4.00 | – | 4.01 | – | 4.00 | 4.00 | 4.00 | – | 3.96 | – | |
| $X_{Alm}$ / $X_{Jd}$ | 0.00 | 0.00 | 0.00 | 0.00 | 0.00 | 0.37 | 0.43 | 0.40 | – | 0.41 | – | 0.42 | – | 0.94 | 0.81 | 0.91 | – | 0.95 | – | |
| $X_{Sps}$ / $X_{Acm}$ | 0.04 | 0.00 | 0.00 | 0.00 | 0.00 | 0.11 | 0.11 | 0.10 | – | 0.09 | – | 0.12 | – | 0.06 | 0.12 | 0.08 | – | 0.00 | – | |
| $X_{Prp}$ / $X_{Aug}$ | 0.00 | 0.00 | 0.00 | 0.00 | 0.00 | 0.51 | 0.47 | 0.50 | – | 0.50 | – | 0.45 | – | 0.00 | 0.06 | 0.00 | – | 0.05 | – | |
| $X_{Grs}$ | 0.91 | 0.93 | 0.01 | 0.95 | 0.01 | | | | | | | | | | | | | | | |
| $X_{Adr}$ | 0.08 | 0.06 | 0.01 | 0.04 | 0.01 | | | | | | | | | | | | | | | |

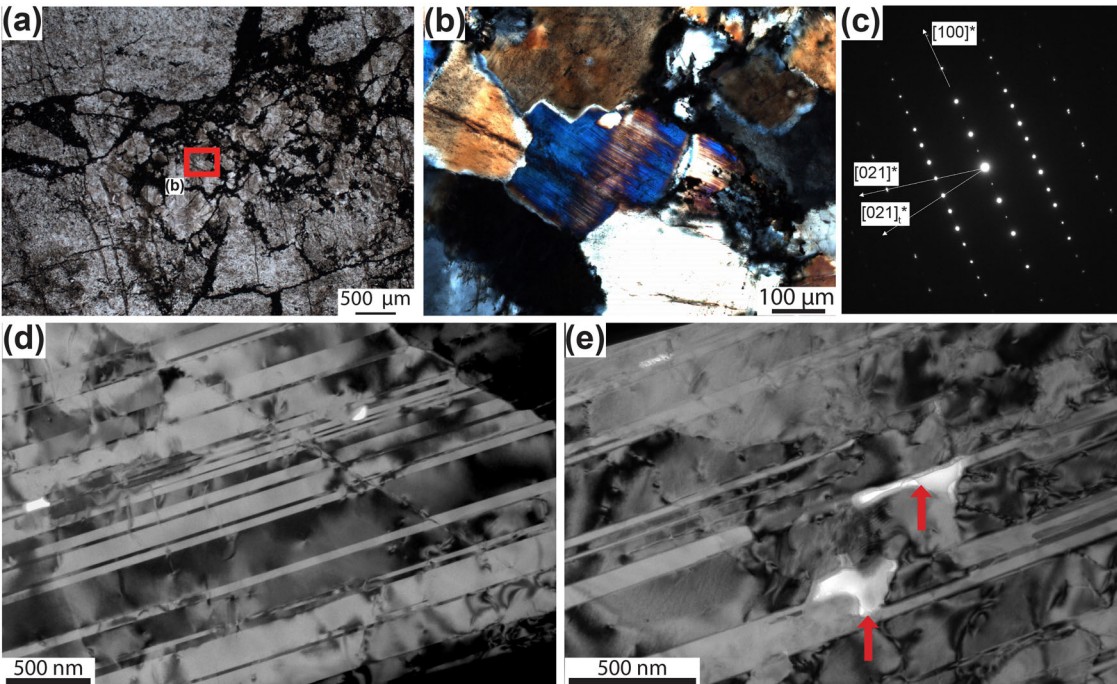

**Fig. 5 | Optical and transmission electron microscopy of twinned omphacite grain. a** Omphacitite clast displaying a pervasive network of fractures coated by graphite. Note the location of the omphacite grain shown in (**b**; thin section photo, plane-polarized light). **b** Detail of twinned omphacite grain on which TEM was performed (thin section photo, crossed-polarized light). **c** Selected area of electron diffraction pattern illustrating mirror law of twins along (100), here with the direction [021]*. **d** TEM bright field image showing numerous planar defects identified as twins on (100). **e** Carbon-rich inclusions along the twins (arrows).

fluid was $CH_4$-$H_2$-rich (Fig. 6). Talcschists and antigorite serpentinites adjacent to the omphacitite are characterized by low permeability and may have represented a permeability barrier for the $CH_4$-rich fluids generated in the underlying carbonated serpentinite. A recent experimental study showed that if serpentinites and talcschists are deformed, permeability increases in the former but not in the latter[49,60]. Consequently, talcschists could act as capping seals until $CH_4$-$H_2$-rich aqueous fluids produced supralithostatic pore pressure values, leading to brecciation of the surrounding rocks and migration of $CH_4$-$H_2$-rich fluids in the newly formed porosity[6,61,62]. Subsequent ductile deformation still occurring at high-pressure conditions and, during the retrograde path, at greenschist facies conditions might be related to subsequent aseismic creep (Supplementary Fig. 4e, f).

Brecciation has often been considered a geological record of seismic failure[63,64], although some caution should be used in the absence of other distinctive textures, such as pseudotachylites[65] or fluidized ultracataclastic veins[66]. In particular, cataclastic products that display strong shape fabrics may also be related to aseismic shearing flow[64]. Noteworthy, in the study samples, there is no evidence of internal organization of the clasts or the matrix producing an anisotropy, such as a foliation, but only of local and successive ductile overprint. Pseudotachylites were not identified due either to their absence or to the fact that they are easily altered and overprinted in geological settings rich in fluids, such as subduction zones[67]. Nonetheless, several authors related the cauliflower structure of garnet to fast growth and seismic activity[54,55]. Moreover, the brecciation of omphacite-rich, strong rock types at subduction zone high-pressure conditions has been in some cases associated with deep seismic ruptures and transiently high pore pressure[56,68,69]. Additionally, dilation breccias similar to the ones described in this study are also found at shallower (crustal) depths in carbonate rocks and have been related to seismic failure[20,57]. Furthermore, it is conceivable that hydrofracturing also involved transient frictional shear deformation, analogous to experiments in serpentine where reaction-induced extension and shear fractures developed simultaneously[70] and to natural conditions where hydrofracturing and frictional slip along phyllosilicate-rich foliations occurred simultaneously[71]. As in the dilation breccias example described above, the studied omphacitite shows evidence of one major brecciation event, possibly because the sealed breccia became harder than the surrounding omphacitite. An alternative interpretation may imply that only a main stage of fluid migration through this rock volume occurred. Additionally, fluids kept open the fracture network until complete sealing, as no significant collapse structures are visible except for the graphite-enriched stylolitic structures between interpenetrated clasts. The fracture network was rapidly sealed at disequilibrium conditions, as evidenced by cauliflower garnet and dendritic/acicular intergrowths of jadeite and omphacite (Fig. 4). Therefore, based on the above features, we suggest that brecciation of omphacitite was plausibly associated with seismic activity at deep forearc conditions.

It is well-established that aqueous fluids can cause hydrofracturing and promote seismic activity in subduction zones[5]. Likewise, it has been shown that carbon-bearing aqueous fluids, such as $CO_2$-rich aqueous fluids, deriving from orogenic degassing can accumulate under geological traps at crustal depths leading to fluid overpressure and the so-called carbo-fracturing[18–22,29]. The term overpressure is generic[37] and literature data point to either sublithostatic or supralithostatic conditions. In the present case study, Mohr-Coulomb diagrams (Fig. 7) point to supralithostatic fluid pressure. To assess the potential of $CH_4$-bearing aqueous fluids for supralithostatic hydrofracturing at subduction zone conditions, we performed thermodynamic modeling in the P-T range of $CH_4$ formation in the Lanzo Massif. Thermodynamic modeling was achieved by computing a pseudosection along the retrograde path of the Lanzo Massif (y-axis; passing through the 2 GPa, 600 °C and 1 GPa, 400 °C conditions) for a COH fluid[72] at carbon saturation conditions. The latter condition is suggested by the precipitation of graphite in both the source area of the $CH_4$-$H_2$-rich fluid –the reduced carbonated serpentinite–, and in the omphacitite breccia matrix (Fig. 1e). Although the petrographic analysis strongly suggests disequilibrium conditions during breccia

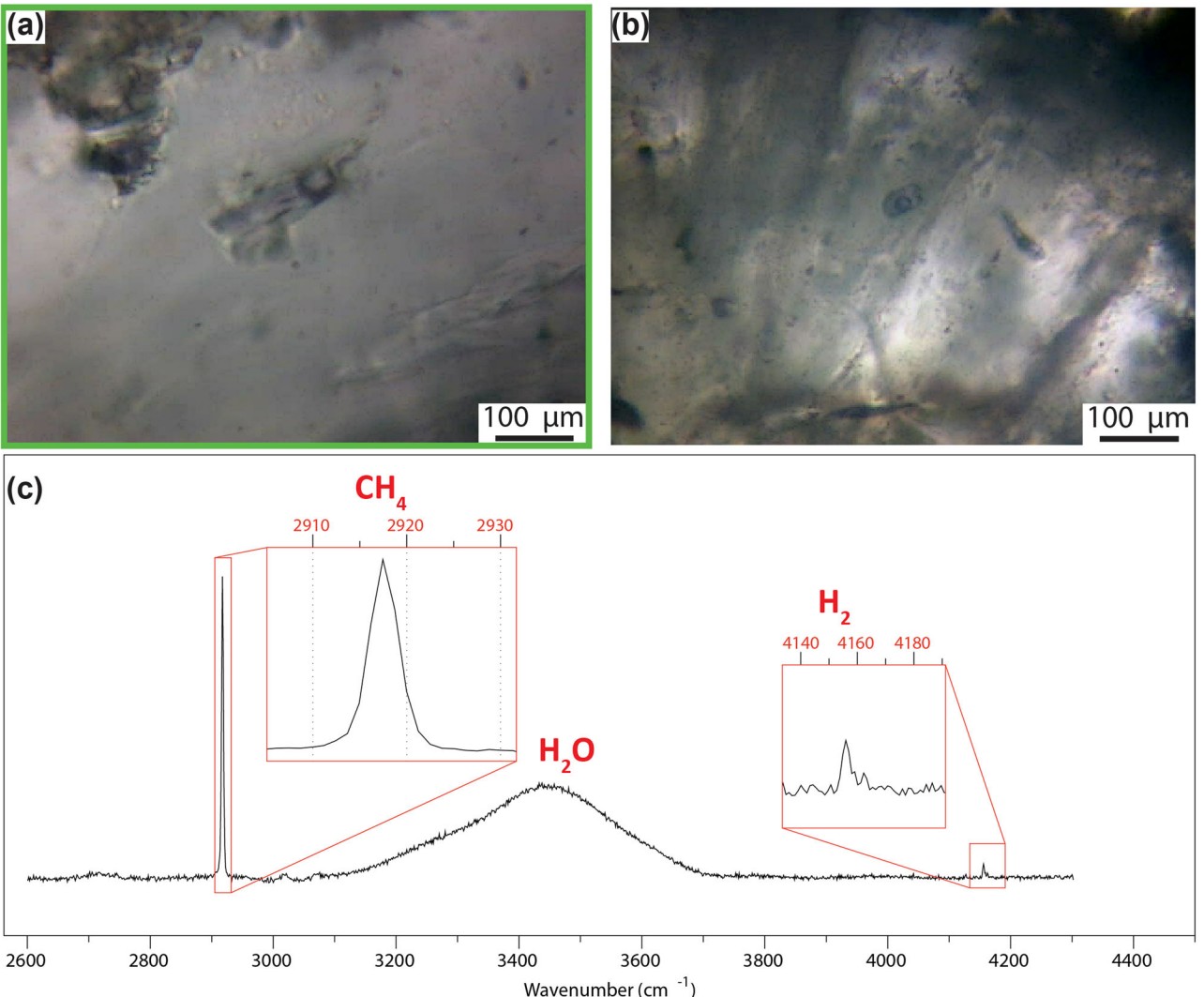

**Fig. 6 | Jadeite-hosted two-phase (liquid-gas) fluid inclusions from the matrix of omphacitite breccias. a, b** Thin section photos, (plane-polarized light).
**c** Representative Raman spectra of $CH_4$–$H_2$–$H_2O$ fluid inclusions.

sealing, the modeling was performed considering fluid-graphite equilibrium in order to expand the potential implication of the modeling to other graphite-bearing rock systems, such as metapelitic rocks. In Fig. 8a, the x-axis refers to the redox state of the graphite-saturated COH fluid [$X_O = nO/(nO + nH + nC)$ in the COH fluid[72]]; For each point along the P-T gradient, the molar volume change between a graphite-saturated COH fluid and pure $H_2O$ was computed. For $X_O = 0.333$, a fluid in equilibrium with graphite contains its maximum amount of water and its minimum amount of carbon, so that the difference relative to pure $H_2O$ is minimized. Moving across the x-axis towards $X_O$ values higher or lower than 0.333, the carbon content in the fluid increases, and the $H_2O$ fraction decreases (Fig. 8b). It can be observed that this increase of carbon in the graphite-saturated COH fluid ($X_O \neq 0.333$) results in a substantial increase in the molar volume of the fluid relative to the water maximum ($X_O = 0.333$). This means that, to reach hydrofracturing conditions, a smaller amount of carbon-bearing fluid is required relative to pure $H_2O$. Because $CH_4$ has a molar volume higher than $H_2O$, a $CH_4$-rich fluid as the one observed in the study rocks ($X_{CH_4} = 0.65 = 0.65$) would reach fluid overpressure conditions suitable for hydrofracturing with a much smaller amount of fluid relative to pure $H_2O$, or even relative to a graphite-saturated fluid at water maximum[30]. For example, Fig. 8a shows that the volume change of a

$CH_4$ rich fluid relative to a $H_2O$ fluid is about 70% for $X_O$ values comparable to those estimated from the natural fluid inclusions.

Figure 8 also shows that the hydrofracturing potential of a graphite-saturated COH fluid is higher for $CH_4$-rich, reduced fluids relative to $CO_2$-rich, oxidized fluids. Additionally, $CH_4$ and $CO_2$ expand more compared to $H_2O$ when the confining pressure is reduced[30,73]. Therefore, $CH_4$ could expand and produce hydrofracturing as soon as it reaches a more porous medium or when the confining pressure drops. Figure 8 also suggests that, if the $CH_4$-rich fluids remained trapped below the talcschist seal during decompression, fluid volume expansion would promote hydrofracturing for even smaller amounts of trapped fluid.

$H_2O$, $H_2$, $CO_2$, and $CH_4$ are immiscible over a wide range of P–T conditions in subduction zones[25–28,74]. Evidence of immiscibility has been described in the carbonated serpentinites representing the source rock[28,75]. Fluid phase separation can occur in immiscible fluids, and $H_2O$ fluid can move much easier compared to $CO_2$-rich fluid due to the higher wetting angles of the latter that preclude a thin film at grain boundaries[24,76,77]. Unfortunately, to the best of our knowledge, no data on the wetting angle of $CH_4$ are available so far, preventing the understanding of its behavior. Finally, seismic activity can enhance the production of effervescing fluid, thus further favoring the migration of carbon-rich fluids[18].

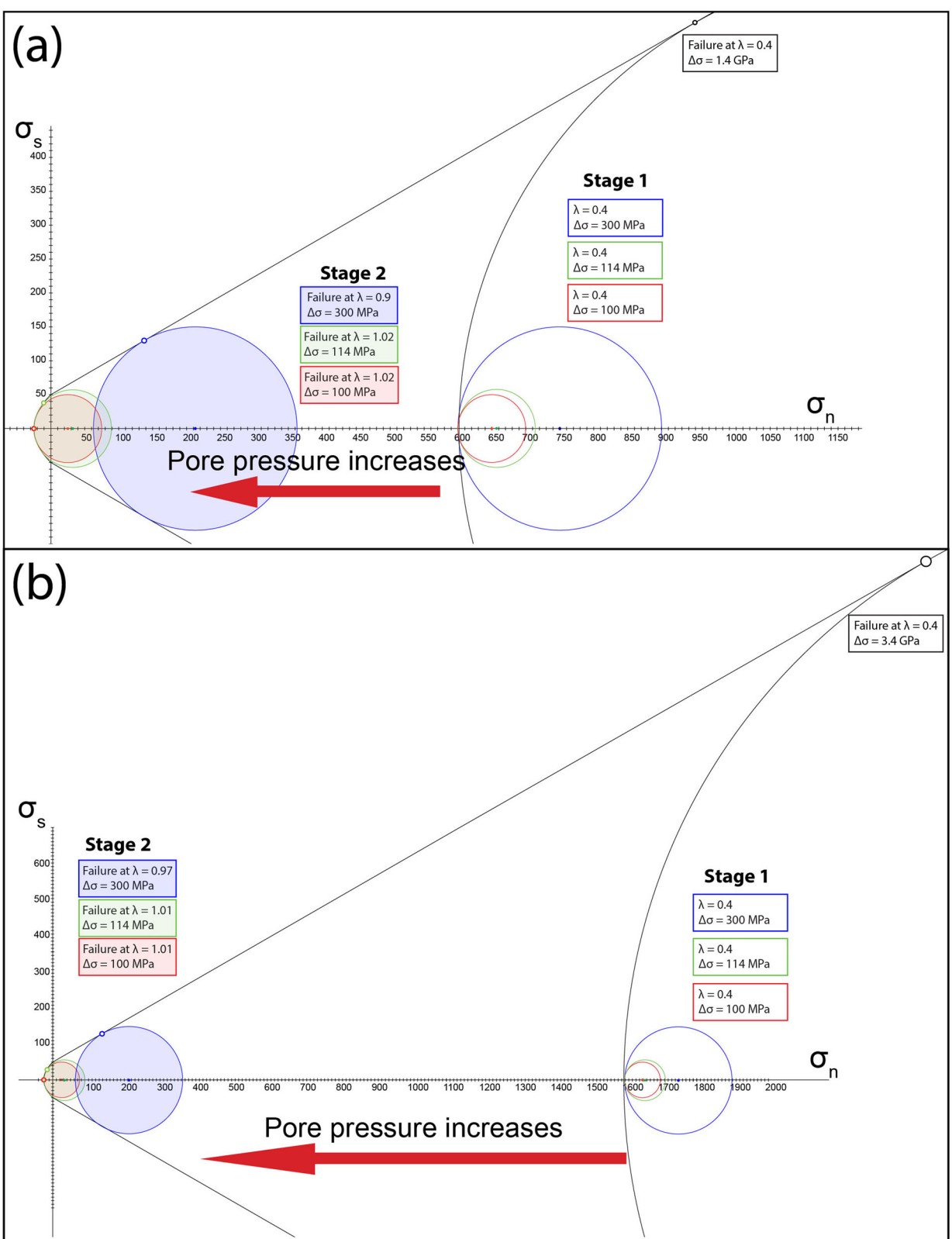

**Fig. 7 | Mohr-Coulomb diagram (normal stress σ$_n$ - shear stress σ$_s$) for the studied omphacitite.** The diagrams were computed for vertical stresses (σ$_V$) of 1 and 2 GPa (**a**, **b**, respectively), corresponding to the minimum compressive stress (σ$_3$; see methods for parameters and assumptions). Failure occurs at Δσ of ~ 1.4 **a** and 3.4 **b** GPa for hydrostatic pore fluid factor values (λ = 0.4); shear failure occurs at Δσ of 300 MPa for sublithostatic λ values, both hybrid and extension failure occur at Δσ of 114 MPa and 100 MPa for supralithostatic λ values.

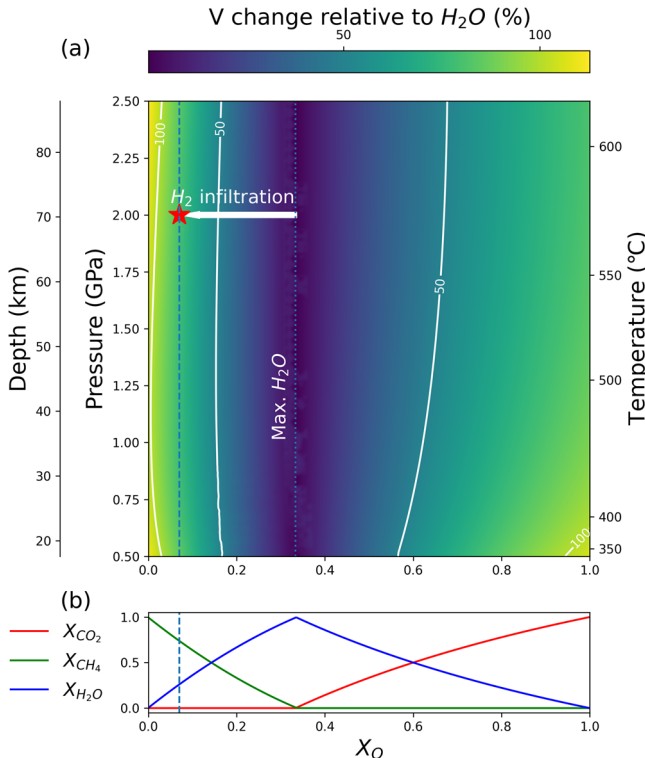

**Fig. 8 | Thermodynamic modeling results. a** Colormap and contours (white lines) of the fluid volume (in % increase) relative to a pure $H_2O$ fluid. Fluid immiscibility at lower P-T conditions is displayed on top of the volume colormap for both $CH_4$-rich fluids (below $X_O = 0.33$) and $CO_2$-rich fluids (above $X_O = 0.33$). The star refers to the $X_O$ value derived from the fluid inclusions (Methods for details about carbon-saturation/undersaturation conditions). **b** Evolution of the fluid composition at 2 GPa and 570 °C in terms of molar fractions of $CO_2$ (red), $CH_4$ (green), and $H_2O$ (blue).

Summarizing, we propose the following stages of deformation of omphacitite layers inside serpentinites of the Lanzo Massif, occurring during Alpine subduction around 50 Ma[36] at eclogite-facies conditions (Fig. 9):

1. At low pore pressure conditions (hydrostatic), brittle failure in omphacitites would require differential stresses of ~1.4 GPa and 3.4 GPa for vertical stresses of 1 and 2 GPa, respectively. These conditions were not attained, but high differential stresses, in the order of some hundreds of MPa, produced mechanical twinning in omphacite. These high differential stresses likely reflect pre-rupture loading.
2. The migration and accumulation of $CH_4$–$H_2$-rich fluids under low permeability talcschists produced pore pressure increase to supralithostatic values, leading to brittle failure and brecciation of the omphacitites. Differential stresses around 100–114 MPa are required for extension failure and hybrid failure, respectively. The high molar volume of $CH_4$ leads to fluid overpressure more easily compared to aqueous fluids. Additionally, immiscibility could also enhance fluid phase separation and accumulation.
3. The fracture network was rapidly sealed at disequilibrium conditions by jadeite, omphacite, garnet and graphite. No evidence of further eclogite-facies brittle failure is recorded, suggesting (i) a hardening of the rock volume, (ii) a single major stage of $CH_4$-$H_2$ migration through the rock volume at peak conditions or (iii) a migration of the rupture in an adjacent volume of undeformed omphacitite. Probably the brecciation was associated with seismic failure.

4. Subsequent local ductile deformation of the breccias occurred still at eclogite-facies conditions and during the retrograde path at greenschist facies conditions, recording aseismic creep.

Concluding, this study suggests that at a depth range between 30 and 80 km in subduction zones the genesis and migration of fluids carrying deep energy sources can lead to supralithostatic pore-fluid pressure and trigger brittle failure in omphacite-rich, mechanically strong rock types similar to eclogite. Modeling results suggest that these $CH_4$-$H_2$-rich fluids can lead to brittle failure much more easily compared to water-dominated fluids, with a $CH_4$-$H_2$-rich fluid having a volume 70% higher than pure $H_2O$ at these conditions (Fig. 8). Immiscibility and phase separation may have favored preferential accumulation of carbon-hydrogen fluids via hydrofracturing. We propose that talcschist and serpentinites acted as low-permeability barriers and seal horizons, thus allowing pore pressure to increase until (supra)lithostatic conditions. We envisage that genesis and migration of $CH_4$-$H_2$-rich aqueous fluids may trigger seismic activity in subduction zones at forearc depths. These processes may play an important role in promoting the migration of deep energy sources from deep source areas towards shallower reservoirs, including the subsurface biosphere where microbial life can take advantage of them through metabolic processes[2–4].

## Methods

### Optical cathodoluminescence (CL)
CL was performed at the University of Bologna (Italy), using a NewTec Scientific Cathodyne motorized optical cathodoluminescence. Images were acquired at 15 kV, ~300 μA/mm² under a pressure of ~0.01 Bar.

### Scanning electron microscopy (SEM) and electron backscatter diffraction (EBSD)
SEM data were acquired with a Zeiss Ultra 55 field emission gun at the Sorbonne Université, Paris (France). High vacuum was used, with an accelerating voltage of 15 kV and a working distance of 7.5 mm. Back-scattered electron (BSE) data were acquired with an Angle Selective Backscattered Detector or an Energy Selective Backscattered Detector.

EBSD data were acquired with a Jeol JSM 6610LV SEM at the Electron Microscopy Centre of the University of Plymouth (United Kingdom). EBSD patterns were collected using a 20 kV accelerating voltage, 18-23 mm working distance, 1 μm step size and a 70° sample tilt. AZtec software (Oxford Instruments) automatically indexed diffraction patterns. Raw maps were processed with HKL Channel 5 (Oxford Instruments), using the noise reduction procedure of[78]. Grains smaller than 3 times the step size were removed from the dataset. The mean angular deviation value was 0.66, the raw indexing rate of 85%, as only omphacite was indexed. Crystallographic orientation data were plotted on pole figures (stereographic projection; upper and lower hemispheres), with X parallel to the stretching lineation and Z parallel to the pole of the foliation. The grain orientation spread maps (GOS maps) were used to display the intensity of internal strain of individual grains. GOS is defined as the average misorientation angle between each pixel in a grain and that grain's average orientation. Grain reference orientation deviation (GROD) map displays the misorientation angle measured at every pixel in a grain with respect to the mean orientation of the grain. Grain size in maps and diagrams was defined as the diameter of the equivalent circle.

### Transmission electron microscopy (TEM)
As described by[79], focused ion beam (FIB) foils were prepared for Transmitted Electron Microscopy (TEM) using a FEI strata DualBeam 235 FIB (at IEMN, Lille). The surface was protected with platinum strip, ~25 μm long and ~2 μm thick. Material on each side of the region of interest was removed by a gallium (Ga) ion beam (30 kV, 7 nA). Then, FIB sections were lifted out and attached onto a copper TEM grid by

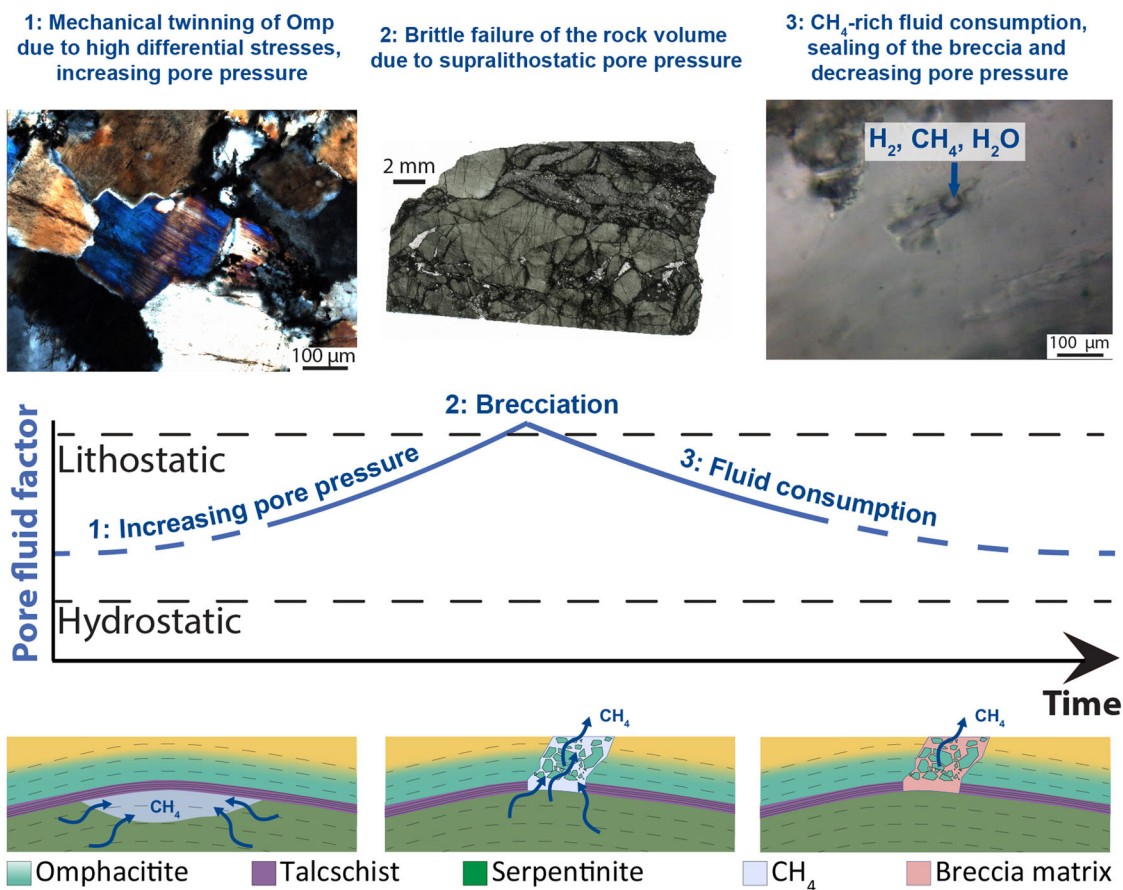

**Fig. 9 | Interpretation of the studied structures related to pore pressure fluctuations.** 1. High differential stresses produced mechanical twinning in omphacite. This stage marks progressive prerupture loading. 2. Brecciation of omphacitites occurred at supralithostatic λ values. 3. Fractures were sealed by jadeite, omphacite, garnet, titanite and graphite growth. Fluid inclusions in omphacite recorded the hydrocarbon-bearing fluids composition. Figure created with Adobe Illustrator CS6 (https://www.adobe.com/products/illustrator.html).

depositing platinum at the contact(s) between the foil and the grid. The section was thinned to ~150 nm using low beam current (1 nA to 300 pA and 100 pA) grazing on both side of the section. Secondary electrons images were taken to control the thinning process. The amorphous material redeposited by the plasma was removed by scanning the foil with a 5 kV ion beam at an angle of 4–7° with the foil surface. FIB sections were studied by TEM using a Thermo Fisher Scientific Tecnai G2-20 (LaB6 filament) operating at 200 kV at the electron microscopy platform of the University of Lille. The study of the microstructure was performed using bright and dark field imaging in conventional TEM mode. Structural information was obtained by selected area electron diffraction (SAED). Semi-quantitative compositions were measured using energy dispersive X-ray spectroscopy (EDXS).

**Electron probe micro-analyser (EPMA)**
As described by[35], EPMA data were acquired at the Camparis Platform, Sorbonne Université, using a Cameca SX Five equipped with five spectrometers. Spot analyses and X-ray maps were acquired with wavelength dispersive spectrometers (WDS). Firstly, spot analyses were acquired for each mineral phase, successively X-ray maps were collected on overlapping areas. Spot analyses were acquired with 15 KeV accelerating voltage, 10 nA specimen current and ~1 μm beam diameter. The following standards were used to measure ten oxide compositions: orthoclase ($Al_2O_3$, $K_2O$), garnet ($SiO_2$, MgO, FeO), albite ($Na_2O$), diopside (CaO), manganese titanate ($TiO_2$, MnO), chromium oxide ($Cr_2O_3$). X-ray maps were acquired with 15 KeV accelerating voltage, 10–100 nA specimen current, dwell times of 100 ms and step size of 1 μm. Ten elements (Si, Ti, Al, Fe, Mn, Mg, Na, Ca, K and Cr) were collected in two series for intensity X-ray maps. Successively, intensity X-ray maps were processed using XMapTools 3.4.1[80], using spot analyses as internal standard to obtain concentration maps of oxide weight percentage.

**Raman spectroscopy**
Raman spectra were acquired with a Renishaw InVIA Reflex microspectrometer at the Laboratoire de Géologie of the Ecole Normale Supérieure, Paris, France. As described by[35], 514 nm laser was used, in polarized mode, delivering 20 mW on the sample, a long-working-distance ×50 Leica objective lens with 0.5 numerical aperture, and 1800 grooves/mm gratings. Analyses were conducted on polished thin sections. A silicon standard was used for calibration. An acquisition time of 5–10 s was used, with 2–4 accumulations per spot. Based on the results of fluid inclusion analysis, the areas of the gaseous molecules were used to estimate the proportion of the main gaseous molecules following the method of[81]. That the gas bubble contains 14 mol% of $H_2$ and 86 mol% of $CH_4$ (no $CO_2$ detected). The method does not allow us to estimate the proportion of $H_2O$.

**Mohr-Coulomb failure diagrams**
The Mohr-Coulomb failure diagrams (normal stress $σ_n$ - shear stress $σ_s$) were plotted using the software MohrPlotter v 3.0[82] available at https://www.rickallmendinger.net/mohrplotter. Pore fluid factor λ was defined as the ratio of fluid pressure to vertical stress $σ_v$[83]. A coefficient

of internal friction (μ) of 0.58, a cohesive strength (C) of 50 MPa, and a tensile strength (T) of 25 MPa were used for omphacite (values from ref. 84). Diagrams were computed assuming an Andersonian stress field and a thrusting regime with the vertical stress $\sigma_v$ corresponding to the minimum compressive stress $\sigma_3$.

### X-ray microscopy (XRM)

X-Ray Microscopy (XRM) was performed using a ZEISS Xradia Versa 610, available at the Research Center on Nanotechnologies Applied to Engineering (CNIS)−Sapienza University of Rome (Italy). This instrument overcomes the limits of traditional X-ray Computed Tomography (CT) featuring a two-stage magnification architecture (Geometric Magnification + Optics) and a high flux X-ray source to produce sub-micron scale resolution images with enhanced contrast. XRM relies on the same X-ray CT approach to collect a set of 2D projections (radiographs) of the specimen at different viewing angles, acquired by rotating the sample and exposing it to the X-ray beam. The resulting projections are then reconstructed using a reconstruction algorithm to obtain a 3D dataset. Here, the Feldkamp−Davis−Kress (FDK) algorithm was used. Absorption contrast tomography was performed to obtain a three-dimensional reconstruction (dataset) of the specimens using a voltage of 50/60 kV, a power of 4.50/6.50 W, a 0.4× objective for scanning the sample in its entirety, and 4× objective for investigating specific regions of interest (ROIs) with higher resolution to provide an isotropic voxel size of 7 μm and 1.3 μm for ROIs, respectively. The sample was 5.2 mm height and 3.9 mm in diameter; ROI 1 1.1 mm and 0.8 mm and ROI2 1.2 mm and 1.1 mm. The exposure time was set between 1 s and 13 s to reach the desired intensity detection and 1601 projections were acquired for low resolution scans whilst 3001/4501 projections were set for higher resolution investigations. The identification of ROIs was conducted using the *Scout-and Zoom* procedure. Then, image processing, analysis, and segmentation were performed using Dragonfly Pro software. High resolution datasets were filtered using a non-local means filter (Kernel Size 9; Smoothing 0.5) to reduce noise from images while preserving sharp edges.

### Thermodynamic modeling

Thermodynamic modeling of fluid inclusion component speciation [e.g., $X_{CH_4}$ $(CH_4/(CH_4 + CO_2 + H_2O + H_2)]$ and redox state ($X_O$ and $fO_2$) was performed using the thermodynamic database of Huizenga[85] and Connolly[72] for the COH system. The measured $CH_4$-$H_2$ proportions in the gas phase derived from the fluid inclusions suggest that the fluid trapped in the fluid inclusions is carbon-undersaturated. The $X_O$ value calculated for this fluid is 0.07 which, at 550 °C and 2 GPa corresponds to an $fO_2$ value about 7 log units below the FMQ buffer. These estimates were obtained under assumption that the fluid inclusions preserved their original $X_O$ value[86]. Post-entrapment respeciation of the fluid inclusions should have moved the fluid towards carbon saturation (Cesare 1995)[86], suggesting that our estimates are conservative with respect to the reduced nature of the fluid. Although the fluid inclusion composition suggests carbon-undersaturated conditions, petrographic evidence (graphite precipitation in the breccia matrix) indicates that the fluid was at least transiently carbon saturated. These fluctuations may be explained by transient pressure variations during the breccia formation, or disequilibrium conditions, as also suggested by the breccia sealing materials.

The thermodynamic modeling for Fig. 8 was done using Perple_X[87] version 7.07.71 and the thermodynamic database of Holland and Powell[88], modified in 2002. The system of components was modified to have C, $O_2$, and $H_2$ using ctransf. The x axis was set between 1 mole of $H_2$ at x = 0 and 1 mole of $O_2$ at x = 1. The graphite saturation was achieved by having one mole of C for the entire pseudosection. The fluid was modeled using the solid solution COH-Fluid.

The molar volume of COH fluid relative to pure $H_2O$ at any P and T along the retrograde path was computed as follows:

$$V\ relative\ to\ H_2O = \frac{V^{P,T}_{COH-fluid}}{V^{P,T}_{H_2O}} \tag{1}$$

Where:

$V^{P,T}_{COH-fluid}$ is the molar volume of COH fluid in equilibrium with graphite (in m³·mol⁻¹) at P and T, $V^{P,T}_{H_2O}$ is the molar volume of pure $H_2O$ at the same P and T conditions in m³·mol⁻¹.

## Data availability

All data generated or analyzed during this study are available within this published article (and its Supplementary information files).

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

## Acknowledgements

This work is part of the project that has received funding from the European Research Council (ERC) under the European Union's Horizon 2020 research and innovation programme (Grant Agreement No. 864045, acronym DeepSeep). A MUR FARE (acronym DRYNK) grant and MUR PRIN2022 (Grant No. 20224YR3AZ; acronym HYDECARB) to A.V.B are also acknowledged. L.M. acknowledges funding from the UK Natural Environment Research Council, grant NE/P001548/1. D. Troadec is thanked for FIB foil preparation. FIB experiments were supported by the French RENATECH network. O. Olivieri is acknowledged for Raman spectra processing. Zeiss Microscopy Italy is thanked for methodological support with X-ray microscopy.

## Author contributions

Conceptualization: F.G., A.V.B. Methodology: F.G., A.V.B., G.S., F.C., H.L., L.M., and M.R. Investigation: F.G., A.V.B., G.S., F.C., H.L., and L.M. Visualization: F.G., A.V.B., G.S., F.C. Writing: F.G., A.V.B. and contribution from L.M., G.S., H.L., F.C., R.C., and M.R.

## Competing interests

The authors declare no competing interests.
