## [Peer Review File · Nature Communications]

REVIEWER COMMENTS

Reviewer #1 (Remarks to the Author):

Giuntoli et al. describe in their paper “Methane-hydrogen-rich fluid migration triggers seismic failure in subduction at forearc depths” spectacular C-bearing omphacite breccias from the Lanzo Massif in the Italian Alps. The deformation record of these samples is characterized by chemical and structural analyses and discussed in the light of coseismic deformation and associated fluid-mediated reactions at depths at which jadeite, garnet and omphacite are stable. Their main conclusion is that between 30-80 km depths in subduction zones CH₄-H₂ fluids at reducing conditions can trigger seismic activity. This conclusion is of significance for understanding seismic activity in subduction zone, material behaviour associated with carbon-rich fluids, and generally the deep carbon cycle. The work is original and stimulating. There are, however, from my point of view a few aspects that would need to be addressed before the manuscript can be published and which are listed as follows:

1. The presentation of data on the carbon phase should be improved: Fig. 1e and Fig 2a are not showing enough detail. Has graphite been characterized by Raman or EBSD? Also, BSE images would be helpful. A better presentation of the phases especially of graphite but also the jadeite-omphacite matrix as well as garnet and titanite would be important. The X-ray maps in Fig. 2b, d as well as Fig. 3 do not show the required detail.
2. An important point of the paper is, as far as I understood, that CH₄-H₂ fluids have been produced at reduced conditions, migrated into the omphacite and lead to precipitation of garnet/omphacite/jadeite and graphite (thus oxidizing conditions are required?) but, the metasomatic/redox reactions are not discussed.
3. The BSE images of Fig. 4 and likewise the X-ray maps of Fig. 2 might instead of an intergrowth of omphacite and jadeite (as stated) rather reflect a metasomatic replacement of omphacite by jadeite where, for example Ca or Fe are exchanged by Na by the metasomatic reactions? It would also have been interesting to see an EBSD map of the matrix, where jadeite and omphacite occur as a mixture. For this discussion, a better characterization of the phases would be required (see also point 1).
4. Fluid inclusions in jadeite are characterized by Raman to contain CH₄ and H₂ (Fig. 6), are there also fluid inclusion in omphacite of the matrix and/or the omphacite “clast”? If not, this would further suggest metasomatic reactions and replacement of omphacite by jadeite instead of new precipitation/intergrowth of all matrix minerals.

5. What exactly is the evidence of overpressure and what is the meaning/ implication? I think the discussion on this topic should be enhanced.

6. The authors commonly write “seismicity” but probably rather mean “seismic activity”. “Seismicity” means the summary of seismic activities in a specific region.

Detailed points by line-numbers are listed as follows:

- Line 23: Please specify what you mean with “deformation zones” - zone of locally higher strain?
- Line 25 (here and elsewhere in the manuscript): “seismic activity” instead of “seismicity”
- Line 25: change to: “effects of carbon-rich aqueous fluid on the strength of deep-seated rocks”
- Line 26: rephrase: e.g. “represent reduced conditions”? Why are they reduced, what are the reactions?
- Line 34: why and how can carbon-rich fluids produce overpressure?
- Line 51: add “on strength”: “negligible effects on strength”
- Line 59: please specify the wetting angle and give a reference
- Line 64: “more reduced” in comparison to?
- Line 71: rephrase: what is the “brecciation at the expenses of omphacite-rich rocks”: do you mean alteration/replacement of omphacite by jadeite, i.e. an Ca/Fe-Na exchange after brecciation?
- Line 107: what is a “dilation breccia structure”? each fracturing is associated with some dilation...
- Line 108: specify “weakly foliated” and further specify “size”, e.g. diameter?
- Line 110: grossular, titanite and graphite are not shown, just by X-ray maps, which is not significant enough. Polarized light micrographs or BSE images might be much more informative.
- Lines 109/110: rephrase: fragments cannot seal, but can for example be cemented by...
- Likewise Line 113 and elsewhere in the manuscript: only fractures can seal, veins are sealed fractures and therefore veins cannot seal...: the veins comprise...
- Line 114: omit “development of an incipient”, this is not needed. Please give reference to a Figure.
- Line 116: what is a “static” foliation?
- Line 118: why do you think it is post brecciation? A discussion would be helpful.
- Line 122: “compenetrated clasts”? I do not really understand what is meant?
- Line 124: Figs. 1a, 2a?
- Line 145: grain diameter?

- Line 185: Again, what are “dilatational breccias”?
- Line 193: Add also reference 48 to: 140-150 MPa 41, 48
- Line 197: add “brittle”: “brittle failure”
- Line 231 small s in similar
- Lines 260, 288: please add some discussion or better explain why “the fluid would reach fluid overpressure conditions”
- Line 333: how was the FIB-microscope operated for the preparation of the TEM foils? Add details on the used dual beam microscope
- Line 600-603: replace “bands” by “layers”, they are 3D-objects...
- Line 624: “intergrown”? or is jadeite replacing omphacite by fluid reactions

Reviewer #2 (Remarks to the Author):

A review of “Methane-rich fluid migration triggers seismic failure in subduction at forearc depth” by F. Guintoli et al.

This paper examines the fracturing of the omphacite layer in serpentinite under eclogite-facies conditions caused by CH₄-rich fluids. The understanding of CH₄-fluid migration and its role in deep subduction zones is limited, despite the authors' previous identification of CH₄-fluids in serpentinite at forearc depths. Through mineralogical and microstructural analyses, as well as thermodynamic considerations of fluid volume and the implementation of a model using the Mohr-Coulomb diagram, the authors propose that the accumulation of CH₄-fluids, resulting from the reduction of carbonates, led to hydrofracturing in the omphacite layer. The topic of energy production, transport, and seismicity in deep subduction zones is timely, and the presence of fractured omphacite with CH₄-bearing fluid inclusions is intriguing. However, it is important to note that certain aspects of hydrofracturing and seismicity in the paper may be speculative in nature.

My main concerns are as follows.

1. The authors proposed that the brecciation of the omphacite layer was caused by the supralithostatic pressure of CH₄-rich fluids. While it is understood that the fluid contained CH₄ and H₂, the authors did not provide the quantitative fraction of CH₄ (e.g., CH₄/(CH₄+H₂O)). Considering that the fractures were filled with silica minerals such as jadeite and garnet, as well as graphite, it suggests that the carbon-bearing fluids are predominantly H₂O-rich. Therefore, the generation, transport, and accumulation of H₂O fluids may be more significant than CH₄. The paper lacks sufficient information about the lithology surrounding the serpentinite, the formation of omphacite layers, and the generation of CH₄ fluids.

Although some of these aspects may have been discussed in the authors' previous paper, the absence of such discussions makes it difficult to determine whether CH₄ fluids played a crucial role in the generation of high fluid pressure and brecciation.

2. The authors suggest a connection between the brecciation and seismicity. However, the evidence of a seismic event is not clearly presented. Furthermore, while it is intriguing, the discussion about the transport of CH₄-rich fluids from eclogite-facies depths to the subsurface biosphere appears too speculative. Are there any specific signatures indicating the presence of CH₄-rich fluids that have passed through the serpentinite bodies?

3. It is unclear whether the intergrowth of jadeite and omphacite can be accurately described as a "dendritic texture."

Overall, although the microstructural and mineralogical analyses are well conducted, unfortunately, it is not enough to show the authors proposal of hydrofracturing induced by CH₄ fluids triggered by seismicity.

Minor comments

P. 26-29 It is important to say how much amount of H₂ and CH₄ are potentially produced in the deep earth, if the authors want to say their impacts on the subsurface biosphere.

P. 54-56 You should say more clearly how CO₂-bearing fluid plays essential roles on the deformation of the lower crust.

P.60-61 It is useful to say that CO₂ metasomatism of hydrated mantle produce fracturing by dehydration and volume change.

Sieber, MJ, Yaxley GM, Hermann J (2020) Investigation of fluid-driven carbonation of a hydrated, forearc mantle wedge using serpentinite cores in high-pressure experiments. *Journal of Petrology* 1-24. doi: 10.1093/petrology/egaa035

Line 69-70 Yes, it is true that how CH₄, H₂ fluids migrate in deep subduction zones is not clear. But the distance between the forearc depths and subsurface is over ~30 km, and thus it is not clear the process observed in this paper is important on such long-distance fluid migration.

Line 73 "rheologically strong rock type"

Not clear. This means the plastic deformation, frictional behavior and brittle fracturing, even for the resistance to the alteration.

Line 84-87 Geological setting should be written more clearly.

At least, the authors need explain lithologies around serpentinites, origin of the ultramafic rocks, timing of carbonation.

Line 87-88 "This process" meaning the circulation of H₂-rich fluids? Explain the relation between and strain localization and fluid channelization and transport.

Line 96 Not clear. This means that carbonate minerals existed with unaltered mantle rock then serpentinization produced the reducing environments resulting in the carbonate minerals decomposed to produce methane?

Line 99 "metamorphic CH₄ production was previously identified"

Explain the occurrences of CH₄ fluid inclusions? The host is carbonate minerals?

Line 100 The timing and formation mechanism of the omphacitite layer should be discussed.

Line 107-113 The field scale occurrence of the breccia is not clear In Fig. 1b. Such fractured zones were developed locally in the omphacitite layer? The zone developed normal to the layer as shown in Fig. 9?

Line 157 Rewrite. "diffuse micro-fracturing"

Line 168 Do you have textural evidence that the twinning planes were strengthened by the precipitation of carbon.

Line 170-172 Acicular grains of omp and jd intergrew in Fig. 4C. But I am not sure whether this texture is called "dendritic", as the crystal orientation is not random.

Line 191-222 Cite figures showing the textural evidence of each event.

Line 192-193 shear stress around 140-150 MPa

Explain how you estimated these values of stress.

Line 210 What is hybrid failure? Shear and extension?

Line 214 You did not show any evidence showing the timing between the formation of talc schist and brecciation of omphacitic layer.

Line 226-227 "there is no evidence of internal organization of the clasts and the matrix" I cannot understand this sentence. Rewrite.

Line 231 "Similar" not capital

Line 223-240 I feel that this is a problem. I don't think that the discussion in this paragraph is not enough to that the brecciation was caused by seismicity.

Line 257 " $0.333 > X_{O_2} > 0.333$ " I cannot understand the meaning of this inequality.

Line 241-268 I understand that molar volume of CH₄ is larger than H₂O and CO₂. But the effect of course depends on CH₄ produced. What is the expected X_{CH_4} ($= CH_4 / (CH_4 + H_2O)$) in fluids?

Line 290-291 The fracture networks were sealed by eclogite-facies silicate minerals. This means that the fluid is aqueous fluids (H₂O-dominated). In this cases, how much CH₄ dissolved in fluids influenced the hydrofracturing? In addition, what is the source of the silicate-forming aqueous fluids? If the H₂O is dominant in fluids, the production and accumulation of H₂O fluids also should be discussed. At the time of fracturing, the ultramafic rocks were completely serpentinized?

Figure 9. Pore fluid factor

I wonder whether lowest fluid pressure can be to hydrostatic even at the 2GPa depth.

Reviewer #3 (Remarks to the Author):

This paper describes samples of a layer of high pressure clinopyroxene (omphacite) rock (given the rock name of omphacitite) in a host rock of serpentinite (hydrated peridotite). The omphacitite is highly fractured, consistent with fractures generated by overpressured fluids, and these fractures are abundantly coated by graphite, thus suggesting that the fluid must have contained methane which was subsequently reduced to carbon by the presence of hydrogen. The hydrogen is assumed to come from the serpentinisation process. The aim of the paper is to go beyond this simple description and demonstrate that such methane-hydrogen-rich fluids are more efficient at generating the seismic fracturing than hydrous fluids.

The geological context is given as the forearc region of a convergent plate boundary with the omphacitite described as a metasomatic layer. To provide a better context for the study, it would have been helpful to have more detail about how this metasomatic layer was formed, even if this was in the Supplementary information. The references (35-37) are unhelpful in this regard. In Figure 1b the omphacitite appears to be a vein within serpentinite. What is being metasomatized, or is it all precipitated from a fluid as implied by the crack-and-seal description of the texture (or does that only refer to veins within the omphacitite?). Given that the focus in the paper is this omphacitite some further details of how it was formed would be helpful for the subsequent interpretation of events.

The deformation twinning is interpreted as the first 'event'. The discussion of the critical resolved shear stress (CRSS) for twin formation is referred to jadeite and diopside, commenting that comparable results were obtained for omphacite. This could be clarified to comment that omphacite is significantly stronger than jadeite or diopside, whereas the text implies that they have similar CRSS. There is also discussion in the literature that whether the omphacite is cation ordered (i.e. $P2/n$) or disordered ($C2/c$) makes a difference to the strength. There is no comment about this in the paper, although diffraction patterns in the TEM could have determined this and perhaps explain why only the grains near the edges of the larger clasts are twinned. The P,T conditions would suggest that the omphacite should be ordered. The TEM describes the twins but no antiphase domains (APDs) are imaged. Perhaps a lost opportunity here.

The jadeite + omphacite intergrowth could also be interpreted as a stable co-existence, according to the Carpenter phase diagram ($P2/n + C2/c$), rather than as dendritic textures interpreted as fast precipitation under nonequilibrium conditions. Again the presence of very fine APD's could have shed some light on this. The interpretation that the intergrowth is dendritic seems to be an important aspect of the overall interpretation (line 239).

Despite these comments, the paper makes a reasonable case that CO₂ bearing fluids may produce greater overpressures than aqueous solutions (higher wetting angles, larger molar volume) and given that the evidence is consistent with fluid-induced fracturing, the conclusion that carbon-bearing fluids could trigger seismic failure is plausible. The paper could be improved with some attention to the issues raised in this review.

Minor comment: References 43 and 47 in the list are identical.

1 **Point-by-point response**

2 We used “track changes” tool to track the changes in the revised manuscript and the red colour to
3 reply to the reviewers’ comments. Please, note that line numbers for the revised manuscript refer to
4 the manuscript file without ‘marked changes’/‘track changes’.

**Reviewer Comments:**

**Reviewer 1**

Giuntoli et al. describe in their paper “Methane-hydrogen-rich fluid migration triggers seismic failure
in subduction at forearc depths” spectacular C-bearing omphacite breccias from the Lanzo Massif
in the Italian Alps. The deformation record of these samples is characterized by chemical and
structural analyses and discussed in the light of coseismic deformation and associated fluid-mediated
reactions at depths at which jadeite, garnet and omphacite are stable. Their main conclusion is that
between 30-80 km depths in subduction zones CH₄-H₂ fluids at reducing conditions can trigger
seismic activity. This conclusion is of significance for understanding seismic activity in subduction
zone, material behaviour associated with carbon-rich fluids, and generally the deep carbon cycle. The
work is original and stimulating. There are, however, from my point of view a few aspects that would
need to be addressed before the manuscript can be published and which are listed as follows:

1. The presentation of data on the carbon phase should be improved: Fig. 1e and Fig 2a are not
showing enough detail. Has graphite been characterized by Raman or EBSD? Also, BSE images
would be helpful. A better presentation of the phases especially of graphite but also the jadeite-
omphacite matrix as well as garnet and titanite would be important. The X-ray maps in Fig. 2b, d as
well as Fig. 3 do not show the required detail.

We strengthened the data presentation of the phases of the matrix adding insets in Fig. 4 and adding
as supplementary material 3 new figures and 2 new detailed microtomography performed in the
matrix of the general one (#1_4a-2). Graphite was not characterised with EBSD, as indexing of
graphite grains was too poor. Graphite was indeed characterized by Raman. Graphite Raman spectra
were collected in different structural sites, such as in vein, along omphacite clast edges, and as
isolated crystals in the breccia matrix. The spectra show some heterogeneity within each structural
context and are consistent with graphitic carbon (Supplementary Fig. 5).

2. An important point of the paper is, as far as I understood, that CH₄-H₂ fluids have been produced
at reduced conditions, migrated into the omphacite and lead to precipitation of
garnet/omphacite/jadeite and graphite (thus oxidizing conditions are required?) but, the
metasomatic/redox reactions are not discussed.

This is an interesting aspect of this complex natural case study. CH₄-H₂ aqueous fluids accumulated
at the top of the serpentinite and then infiltrated the omphacite during the brecciation event. That is
correct. At least part of the breccia matrix may have formed through the reprecipitation of the
precursor omphacite after intense dissolution as evidenced by the presence of stylolites. The
abundance of a generally fluid immobile element such as Al in the matrix assemblage (jadeite,
grossular), supports this hypothesis. Although a detailed mass balance was not done –and would be
challenging owing to the strong heterogeneity of the matrix composition–, the matrix assemblage
appears enriched in Na relative to the precursor omphacite. This may be potentially related to fluid
immiscibility in the close ophicarbonates, which may have enriched the salinity of the aqueous fluid

leading to Na-rich fluid compositions. The interaction with Al-Si-rich rocks during the brecciation of
the omphacite may have favoured the formation of jadeite-rich assemblages. Although this
interpretation is plausible, the fluid inclusions in jadeite do not show evidence for high salinity, such
as daughter halite.

Although these hypotheses may be plausible, their incorporation in the manuscript would require
more text and more references (at least 10 based on a quick survey), and likely more figures. As the
core of this manuscript is structural, and the geochemical modelling can be reasonably simplified to
a COH system, we think that this aspect is beyond the scope of the study and opted for keeping the
petrological modelling of the omphacite for a future contribution.

As for the mechanisms/conditions of graphite precipitation, besides oxidation of a reduced fluid, P
fluctuations may also alter the saturation point of a C-bearing fluid. Here we think that P fluctuations
during the brecciation event may have favoured graphite precipitation. This is not discussed in the
text because other mechanisms, such as kinetics, catalysis and others, may control graphite
precipitation. This question is beyond the scope of this study.

3. The BSE images of Fig. 4 and likewise the X-ray maps of Fig. 2 might instead of an intergrowth
of omphacite and jadeite (as stated) rather reflect a metasomatic replacement of omphacite by jadeite
where, for example Ca or Fe are exchanged by Na by the metasomatic reactions? It would also have
been interesting to see an EBSD map of the matrix, where jadeite and omphacite occur as a mixture.
For this discussion, a better characterization of the phases would be required (see also point 1).

Fair point. If the reviewer refers to the clasts of the preexisting rock, we do not observe any
metasomatic replacement, besides some overgrowth rims along the edges. If the reviewer refers to
the images shown in Figure 4c-h (of the new Fig. 4), those are from the matrix sealing the breccia. In
this case, replacement also had been one of our working hypotheses in the beginning. Finally, we
concluded that co-precipitation was the most likely interpretation. We do not see evidence of

metasomatic replacement textures in the matrix (e.g. lobate edges and peninsular features, evidence
of transient porosity, see e.g. Putnis, A. (2009). Mineral replacement reactions. *Reviews in*
*Mineralogy and Geochemistry*, 70(1), 87–124. <https://doi.org/10.2138/rmg.2009.70>). Instead, we
observe sharp contacts and intergrowths between jadeite and omphacite, suggesting fast and coeval
growth of those two minerals. Mutual apparent cross-cutting relationships are observed, casting
doubts on a simple replacement process (Fig. 4 and Fig. S7). Additionally, rapid growth is suggested
also by the presence of garnet with a cauliflower texture intergrown with jadeite in the matrix (e.g.
*Altenberger et al. (2013). A seismogenic zone in the deep crust indicated by pseudotachylytes and*
*ultramylonites in granulite-facies rocks of Calabria (Southern Italy). Contributions to Mineralogy*
*and Petrology*, 166, 975–994.; *Clerc, A., Renard, F., Austrheim, H., Jamtveit, B. (2018). Spatial and*
*size distributions of garnets grown in a pseudotachylyte generated during a lower crust earthquake.*
*Tectonophysics* 733, 159-170; *Incel et al. 2020. Evolution of brittle structures in plagioclase-rich*
*rocks at high-pressure and high-temperature conditions—Linking laboratory results to field*
*observations. Geochemistry, Geophysics, Geosystems*, 21(8), e2020GC009028.; *Mancktelow et al.*
*2022. Time-Lapse Record of an Earthquake in the Dry Felsic Lower Continental Crust Preserved in*
*a Pseudotachylyte-Bearing Fault. Journal of Geophysical Research: Solid Earth*, 127(4),
e2021JB022878). Noteworthy, the cited reference found a link between the cauliflower structure of
garnet and fast growth related to seismic activity. Finally, cauliflower garnet was also related to rapid
growth under chemical disequilibrium (*Wilbur, D. E., & Ague, J. J. (2006). Chemical disequilibrium*
*during garnet growth: Monte Carlo simulations of natural crystal morphologies. Geology*, 34(8),
689–692). We added this part in the revised manuscript, for sake of clarity. We also implemented the
characterization of jadeite and omphacite in the matrix (please, check our reply to comment #1).

Regarding acquiring an EBSD of the matrix, this is challenging owing to the presence of finely
dispersed graphite that poses a problem for indexing of the diffraction patterns. As a matter of facts,
this is also evidenced by the black (non-indexed) area of Fig.2e,f.

4. Fluid inclusions in jadeite are characterized by Raman to contain CH₄ and H₂ (Fig. 6), are there
also fluid inclusion in omphacite of the matrix and/or the omphacite “clast”? If not, this would further
suggest metasomatic reactions and replacement of omphacite by jadeite instead of new
precipitation/intergrowth of all matrix minerals.

We see what the reviewer means. Measurable fluid inclusions were found in the jadeite located in the
matrix. The fluid inclusions in large jadeite crystals from crack-seal veins are primary (large, isolated
fluid inclusions inside the jadeite crystals). Omphacite and jadeite intergrowth are too fine grained
and cloudy in thin section. We could not identify fluid inclusions in them. Omphacite clasts, instead,
show secondary trails of very tiny fluid inclusions that could not be measured by Raman. Overall,
these features suggest that CH₄-H₂-rich aqueous fluids were present during the brecciation of the
omphacite (secondary trails) and the precipitation of the matrix sealing the breccia (primary
inclusions).

For the replacement part of the question, please see the above reply.

5. What exactly is the evidence of overpressure and what is the meaning/ implication? I think the
discussion on this topic should be enhanced.

For the sake of clarity, we refer to fluid overpressure and not tectonic overpressure. So, if the reviewer
refers to tectonic overpressure, we do not argue for any tectonic overpressure. If the reviewer refers
to fluid overpressure, to reach fracturing at these depths, either extremely high differential stresses
(on the order of GPa) or high pore fluid pressure are needed (e.g., see the discussion regarding a
similar problem in blueschist rocks in Molli et al., 2017, Solid Earth doi:10.5194/se-8-767-2017). As
only minor collapse is documented by the rock microstructures (represented by the graphite-rich
stylolites) and we have evidence of high pore fluid pressure, high pore fluid pressure is the most likely
interpretation for the brecciation, with pore fluid pressure increasing until it overcame the confining
pressure and reached the failure envelop. We expanded the discussion along these lines.

6. The authors commonly write “seismicity” but probably rather mean “seismic activity”.
“Seismicity” means the summary of seismic activities in a specific region.

Accepted and implemented in the revised text.

Detailed points by line-numbers are listed as follows:

- Line 23: Please specify what you mean with “deformation zones” - zone of locally higher strain?

Correct, changed in “shear zones”.

- Line 25 (here and elsewhere in the manuscript): “seismic activity” instead of “seismicity”

Done, thank you.

- Line 25: change to: “effects of carbon-rich aqueous fluid on the strength of deep-seated rocks”

Done, thank you.

- Line 26: rephrase: e.g. “represent reduced conditions”? Why are they reduced, what are the
reactions?

We prefer to state that the fluid is reduced to avoid implications on the equilibrium (fluid rock
equilibrium) or setting (e.g. mantle...). Here we refer to the fluid only, based on fluid inclusions
analysis.

These fluids are reduced because they carry reduced molecules, such as H₂ and CH₄.

- Line 34: why and how can carbon-rich fluids produce overpressure?

Carbon-rich fluids can produce fluid overpressure in the same way of aqueous fluids, by decreasing
the normal stress acting on a surface (effective normal stress = normal stress- fluid pressure e.g. Secor
1965). The point we want to make here is that, based on the thermodynamic calculations presented
in the manuscript (Fig. 8) that consider the larger molar volumes of carbon-rich aqueous fluids

relative to pure water, CH₄-H₂-rich fluids can lead to brittle failure much more easily compared to
water-dominated fluids, with a CH₄-H₂-rich fluid having a volume 70% higher than pure H₂O at these
conditions.

- Line 51: add “on strength”: “negligible effects on strength”

Agreed.

- Line 59: please specify the wetting angle and give a reference

We specified those values in the revised version. The reference is Watson et al., 1987 (ref. 21 of the
submitted version). We moved it to make it clearer for reader.

- Line 64: “more reduced” in comparison to?

Compared to CO₂ rich aqueous fluids. We moved the comma, to make the sentence clearer.

- Line 71: rephrase: what is the “brecciation at the expenses of omphacite-rich rocks”: do you mean
alteration/replacement of omphacite by jadeite, i.e. an Ca/Fe-Na exchange after brecciation?

No, we meant brecciation of omphacite-rich rocks. Apologies for the unclear text. Modified in the
revised text.

- Line 107: what is a “dilation breccia structure”? each fracturing is associated with some dilation...

Following the definition of Tarasewicz et al., 2005 <https://doi.org/10.1130/B25568.1>: “dilation
breccia can be defined as any fragmented rock in which there has been a net volume increase during
formation.

- Line 108: specify “weakly foliated” and further specify “size”, e.g. diameter?

We added two new figures to better show this foliation in the supplement and we specified in the
EBSD method of the revised manuscript that the grain size was defined as the diameter of the
equivalent circle. We changed the term here.

- Line 110: grossular, titanite and graphite are not shown, just by X-ray maps, which is not significant
enough. Polarized light micrographs or BSE images might be much more informative.

**We added more material. Please, refer to our response to comment 1.**

- Lines 109/110: rephrase: fragments cannot seal, but can for example be cemented by...

**Rephrased.**

- Likewise Line 113 and elsewhere in the manuscript: only fractures can seal, veins are sealed
fractures and therefore veins cannot seal.: the veins comprise...

**Rephrased.**

- Line 114: omit “development of an incipient”, this is not needed. Please give reference to a Figure.

**Ok, modified as suggested.**

- Line 116: what is a “static” foliation?

**Deleted static.**

- Line 118: why do you think it is post brecciation? A discussion would be helpful.

**Because it overprints the breccia and the minerals that mark it are characteristic of much shallower**
**(retrograde greenschist facies) metamorphic conditions.**

- Line 122: “compenetrated clasts”? I do not really understand what is meant?

**Modified the term in “interpenetrated”, meaning that the two clasts penetrate mutually with an**
**interface between the clasts similar to a stylolite.**

- Line 124: Figs. 1a, 2a?

No, the original definition “Supplementary Figs. 1,2” is correct, as this refers to figures present in the
supplementary file of the submitted version.

- Line 145: grain diameter?

Correct, modified.

- Line 185: Again, what are “dilatational breccias”?

Please, see our reply to comment to line 107.

- Line 193: Add also reference 48 to: 140-150 MPa 41, 48

Implemented.

- Line 197: add “brittle”: “brittle failure”

Done.

- Line 231 small s in similar

Done.

- Lines 260, 288: please add some discussion or better explain why “the fluid would reach fluid
overpressure conditions”

Please, see our reply to comment n#5 and to comment to line 34 above.

- Line 333: how was the FIB-microscope operated for the preparation of the TEM foils? Add details
on the used dual beam microscope

FIB foils were prepared on microfossils with a FEI strata DualBeam 235 FIB (at IEMN, Lille) for
Transmitted Electron Microscopy. The top surface of each region of interest was protected with
platinum strip $\sim 25 \mu\text{m}$ long and $\sim 2 \mu\text{m}$ thick. Material on each side of the region of interest was
removed by a gallium (Ga) ion beam (30 kV, 7 nA). Then, FIB sections were lifted out and attached

onto a copper TEM grid by depositing platinum at the contact(s) between the foil and the grid. The
section was thinned to ~150 nm using low beam current (1 nA to 300 pA and 100 pA) grazing on
each side of the section. SE images were taken to control the thinning process. Finally, the plasma-
redeposited amorphous material was removed by scanning the foil with a 5 kV ion beam at an angle
of 4–7° with the foil surface.

We added this details in the method section, we thank the reviewer for his request that improved the
clarity of sample preparation.

- Line 600-603: replace “bands” by “layers”, they are 3D-objects...

Ok, done.

- Line 624: “intergrown”? or is jadeite replacing omphacite by fluid reactions

Intergrown is correct, please, see our reply to general comment 3.

**Reviewer 2**

A review of “Methane-rich fluid migration triggers seismic failure in subduction at forearc depth” by
F. Guintoli et al.

This paper examines the fracturing of the omphacitite layer in serpentinite under eclogite-facies
conditions caused by CH₄-rich fluids. The understanding of CH₄-fluid migration and its role in deep
subduction zones is limited, despite the authors' previous identification of CH₄-fluids in serpentinite
at forearc depths. Through mineralogical and microstructural analyses, as well as thermodynamic
considerations of fluid volume and the implementation of a model using the Mohr-Coulomb diagram,
the authors propose that the accumulation of CH₄-fluids, resulting from the reduction of carbonates,
led to hydrofracturing in the omphacitite layer. The topic of energy production, transport, and
seismicity in deep subduction zones is timely, and the presence of fractured omphacitite with CH₄-

bearing fluid inclusions is intriguing. However, it is important to note that certain aspects of
hydrofracturing and seismicity in the paper may be speculative in nature.

My main concerns are as follows.

1. The authors proposed that the brecciation of the omphacite layer was caused by the supralithostatic
pressure of CH₄-rich fluids. While it is understood that the fluid contained CH₄ and H₂, the authors
did not provide the quantitative fraction of CH₄ (e.g., CH₄/(CH₄+H₂O)). Considering that the
fractures were filled with silica minerals such as jadeite and garnet, as well as graphite, it suggests
that the carbon-bearing fluids are predominantly H₂O-rich. Therefore, the generation, transport, and
accumulation of H₂O fluids may be more significant than CH₄.

The reviewer is right, the fluid was a CH₄-H₂-rich aqueous fluid. The fluid inclusions analysis by
Raman clearly shows that (Figure 6 of the submitted manuscript). We clarified that throughout the
manuscript. Although a pure CH₄-H₂ fluid would greatly reduce the amount of fluid capable to reach
overpressure compared to a pure aqueous fluid, we are not suggesting such a scenario. Figure 8 shows
the hydrofracturing potential of COH aqueous fluids, from water-maximum conditions ($X_O = 1/3$), to
carbon enriched (CH₄-rich to the left; CO₂-rich to the right) conditions. The figure shows that, even
in the presence of aqueous fluids, the addition of carbon greatly enhances the hydrofracturing
potential. The Figure was modified with another panel to show the molar fractions of H₂O, CH₄, and
CO₂ across a horizontal profile at 2 GPa.

As for the proportion of CH₄ in the aqueous fluid, Raman spectra cannot be used to derive the
CH₄/H₂O ratio. However, we used the obtained Raman spectra to estimate the proportions of CH₄
and H₂ in the gaseous part of the fluid, which is 86% CH₄ and 14% H₂ (no other gaseous molecules
detected). Thanks to a software developed in our group (Boutier et al., submitted), we estimated the
conditions for which this proportions are possible within a COH ternary system at 550 °C and 2 GPa.
We found that, at these conditions, the measured CH₄/H₂ ratio is possible for a carbon undersaturated

fluid at $X_{\text{O}} \simeq 0.07$, which corresponds to $\Delta\text{FMQ-7}$ (previous work estimated $\Delta\text{FMQ-6}$ for the CH_4 -
forming event). At the estimated conditions, the $X_{\text{H}_2\text{O}}$ is 0.25, whereas the X_{CH_4} is 0.65 ($X_{\text{H}_2} =$
0.1). So, CH_4 was dominant in this aqueous fluid. The carbon-undersaturated nature of this fluid in a
graphite-bearing vein system may be explained by pressure fluctuations during the breccia formation
process, or by disequilibrium conditions, as suggested by the vein mineralogy. Because these
considerations do not take into account fluid inclusion respeciation during exhumation, the revised
manuscript now presents some of these results as a qualitative assessment.

The paper lacks sufficient information about the lithology surrounding the serpentinite, the formation
of omphacite layers, and the generation of CH_4 fluids. Although some of these aspects may have
been discussed in the authors' previous paper, the absence of such discussions makes it difficult to
determine whether CH_4 fluids played a crucial role in the generation of high fluid pressure and
brecciation.

The geological setting has been considerably rewritten. The CH_4 presence is testified by fluid
inclusions in the breccia matrix of the studied rock. Here we quantify the effect of CH_4 -bearing
aqueous fluids compared to pure H_2O fluids. The results and Fig. 8 show that the presence of CH_4 in
the aqueous fluid decreases the amount of fluid required to reach overpressure by several tens percent.

2. The authors suggest a connection between the brecciation and seismicity. However, the evidence
of a seismic event is not clearly presented.

In this article, we speculate that the brecciation could be related to seismic activity occurring during
subduction at forearc depths, based on the geological record of mixed brittle-viscous deformation
occurred at P-T-fluids conditions consistent with the depth of seismological and geodetic record of
seismic fault behaviour in the subduction zone. This is evidenced already at the end of the abstract of
the submitted article, where we wrote that “may trigger seismicity”. Nonetheless, taking into account

this issue raised by the reviewer, we decided to change the title of the manuscript in “may trigger...”,
to further strengthen that this is a proposition.

We would like to reiterate what are the lines of evidence that allow us to propose a link between
brecciation and seismic activity:

- • The breccia formed at >1GPa and 400°C;
- • High pore fluid pressure is needed to reach the brittle failure (see reply to Reviewer 1 comment
5 and specific comment to line 34); similar occurrences of hydrofracturing at eclogite facies
conditions in subduction zones were related to seismic events (e.g., Angiboust et al., *Geology*
2012 doi:10.1130/G32925.1);
- • The breccia matrix has microstructural evidence of fast mineral growth under chemical
disequilibrium;
- • The dilation breccia formation produces a volumetric deformation that, for strain
compatibility, is likely to trigger simultaneous shear deformation as shown in natural and
experimental studies (e.g., Menegon & Fagereng, 2021 *Geology*
<https://doi.org/10.1130/G49012.1>; Zheng et al., 2019 *Journal of Geophysical Research: Solid*
*Earth* <https://doi.org/10.1029/2018JB017008>).
- • The area affected by the brecciation, which gives an estimate of the size of the potential
rupture area of a seismic event, is at least several metres wide (this information was
implemented in the revised manuscript).

Therefore, based on the literature data and interpretation reported in the discussion (and expanded
based on the Reviewers' comments) we are confident that the link between brecciation caused by
high pore fluid pressure of a methane-rich fluid and seismic activity can reasonably be suggested.
Please, see also reply to relevant specific comments. We thank the reviewer to allow us to clarify this
point.

Finally, we want to stress that what discussed above does not undermine the significance of this
article, as the relevance of this work is the brecciation triggered at forearc depths by a carbon rich
fluid in eclogite-like rock types.

Furthermore, while it is intriguing, the discussion about the transport of CH₄-rich fluids from
eclogite-facies depths to the subsurface biosphere appears too speculative.

We agree with the reviewer, this is a big jump. However, considering the current targets in the
understanding of how deep Earth habitability has evolved, we think that this large-scale perspective
must be highlighted as a long-term challenge. This would also expand the potential audience of this
study to the microbiology community, which is currently reconsidering deep abiotic CH₄ as a key
energy source for deep microbial life (Colman et al 2017 PNAS; Perino et al 2019 Frontiers in
Microbiology; Plumper et al 2017 PNAS). Our study represents a tile of the puzzle.

Are there any specific signatures indicating the presence of CH₄-rich fluids that have passed through
the serpentinite bodies?

We clarified this in the revised geological setting at lines 111-121 of the track-changes revised
manuscript. These results are reported in the cited article of Vitale et al., 2017 Nat Comm.

3. It is unclear whether the intergrowth of jadeite and omphacite can be accurately described as a
"dendritic texture."

We used the descriptive definition of "dendritic" found in Vernon2008 textbook "A Practical Guide
to Rock Microstructure": "*Dendritic: Describes the shape of single crystals with spiky or branched
habit precipitated from fluid at conditions of strong supersaturation*". We implemented the term
dendrite/acicular in the text.

We added 2 new figures (S6 and S7) in the supplementary to further document those structures. In
particular, we added 2 insets in Fig. 4 that show the cauliflower structure of garnet crystallizing in

the matrix interfingered with jadeite, suggesting fast growth of these minerals under chemical
disequilibrium (see also references and reply to comment n°3 of reviewer 1). Accordingly, we added
those relevant references and expanded the discussion of the revised manuscript.

Overall, although the microstructural and mineralogical analyses are well conducted, unfortunately,
it is not enough to show the authors proposal of hydrofracturing induced by CH₄ fluids triggered by
seismicity.

We think that the reviewer meant “hydrofracturing induced by CH₄ fluids triggered by seismicity.”

We already made the point that we only suggest as a possibility that CH₄-bearing aqueous fluid may
have triggered seismicity. Please, refer to the above reply to comment #2.

We want to reiterate that the novelty of the article is that due to its composition, the fluid can reach
more easily fluid overpressure and trigger fracturing of stiff rock types (see reply to line 34 of
reviewer #1). In the samples presented here, such fracturing occurred at P-T-fluids conditions
consistent with the typical seismogenic environment in the subduction zone.

Minor comments

P. 26-29 It is important to say how much amount of H₂ and CH₄ are potentially produced in the deep
earth, if the authors want to say their impacts on the subsurface biosphere.

The text was revised. However, estimates of the fluxes of CH₄ and H₂ from the deep Earth are still
far to be established, and only a few published articles exist (Vitale Brovarone et al., 2017, 2020 Nat
Comm; Zhang et al. 2023 National Science Reviews).

P. 54-56 You should say more clearly how CO₂-bearing fluid plays essential roles on the deformation
of the lower crust.

We clarify the sentence, as requested. Please, see lines 60-77 of the track-changes revised manuscript.

P.60-61 It is useful to say that CO₂ metasomatism of hydrated mantle produce fracturing by
dehydration and volume change. Sieber, MJ, Yaxley GM, Hermann J (2020) Investigation of fluid-
driven carbonation of a hydrated, forearc mantle wedge using serpentinite cores in high-pressure
experiments. Journal of Petrology 1-24. doi: 10.1093/petrology/egaa035

**We added this effect and the relevant reference, thank you for the suggestion.**

Line 69-70 Yes, it is true that how CH₄, H₂ fluids migrate in deep subduction zones is not clear. But
the distance between the forearc depths and subsurface is over ~30 km, and thus it is not clear the
process observed in this paper is important on such long-distance fluid migration.

**We fully agree with the reviewer. However, clarify how the transfer of energy sources from the deep
earth to the surface (or even the deep biosphere) is a long-lasting debate (controversial work by the
Russian-Ukrainian school and T. Gold “Gold, T. The deep, hot biosphere. Proc. Natl Acad. Sci. USA
89, 6045–6049 (1992)”) and a long-term challenge. Once the source areas are identified (like in the
present case study), it becomes mandatory to assess how these fluids can escape from them, and how
they can propagate through the surrounding rocks. This is a first, mandatory step.**

Line 73 “rheologically strong rock type” Not clear. This means the plastic deformation, frictional
behavior and brittle fracturing, even for the resistance to the alteration.

**Correct, we use this general term to point out the important contrast of competence between
omphacite and surrounding rock-types, thus stiff, rigid layers embedded in a weak, viscous matrix.**

Line 84-87 Geological setting should be written more clearly. At least, the authors need explain
lithologies around serpentinites, origin of the ultramafic rocks, timing of carbonation.

**We added what requested, also in line with reviewer 1.**

Line 87-88 “This process” meaning the circulation of H₂-rich fluids? Explain the relation between
and strain localization and fluid channelization and transport.

“The reduction of carbonate rocks”, specified in the revised text.

Line 96 Not clear. This means that carbonate minerals existed with unaltered mantle rock then
serpentinization produced the reducing environments resulting in the carbonate minerals decomposed
to produce methane?

In the Lanzo Massif, carbonated serpentinites are interpreted to have formed as a result of
hydrothermal alteration of the ultramafic body at the seafloor prior to Alpine subduction (30.
Lagabrielle, Y., Fudral, S. & Kienast, J.-R. the oceanic cover of the Lanzo peridotite body (Western
Italian Alps): lithostratigraphic and petrological evidence. *Geodin. Acta* 4, 43–55 (1990).).

More recent work (Debret et al 2013) showed that the massif also experiences serpentinization in the
subduction zone, and (Vitale Brovarone et al 2017) that this process produced H₂-rich fluids capable
to reduce preexisting carbonated serpentinites. We implemented some of this information in the
geological setting.

Line 99 “metamorphic CH₄ production was previously identified” Explain the occurrences of CH₄
fluid inclusions? The host is carbonate minerals?

We modified the introduction, as requested. Specifically, the CH₄ inclusions documented in the Vitale
Brovarone et al., 2017 *Nat Comm* and Giuntoli et al., 2020 *Sci Rep* were in carbonates.

Line 100 The timing and formation mechanism of the omphacitite layer should be discussed.

The omphacitite layer is considered to represent a metasomatized tectonic slice of continental crust
(see also our reply to general comment of Reviewer 3 below). The omphacitite represent the starting
material of our work. The timing of the process is beyond the scope of the present article and does
not influence our findings.

Line 107-113 The field scale occurrence of the breccia is not clear In Fig. 1b. Such fractured zones
were developed locally in the omphacitite layer? The zone developed normal to the layer as shown
in Fig. 9?

The brecciated layer is over a thickness of >2 metres and a length of >5 metres. We implemented this
information in the revised text.

Line 157 Rewrite. “diffuse micro-fracturing”

Accepted.

Line 168 Do you have textural evidence that the twinning planes were strengthened by the precipitation
of carbon.

No, we meant “supported”. We modified the sentence for clarity.

Line 170-172 Acicular grains of omp and jd intergrew in Fig. 4C. But I am not sure whether this
texture is called “dendritic”, as the crystal orientation is not random.

Please, refer to reply to comment 3 above.

Line 191-222 Cite figures showing the textural evidence of each event.

Implemented in the revised text, as requested.

Line 192-193 shear stress around 140-150 MPa. Explain how you estimated these values of stress.

This is reported in the cited reference. We changed the sentence to clarify this point. Note that this is
the critical resolved shear stress, the corresponding differential stresses are on the order of twice as
much.

Line 210 What is hybrid failure? Shear and extension?

Correct, this is a terminology commonly used and accepted in the structural geology community (e.g.
Fossen 2010 textbook: “Combinations of shear (Mode II or III) fractures and tension (Mode I)
fractures are called hybrid fractures or hybrid cracks.”).

Line 214 You did not show any evidence showing the timing between the formation of talcschist and
brecciation of omphacitic layer.

As we wrote in the geological setting, talcschists, serpentinites and omphacitite layers are parallel to
the trend of the main foliation. As the brecciation of the omphacitite postdate the foliation visible in
the omphacitite clasts, the brecciation also postdates the formation of the talcschists.

Line 226-227 “there is no evidence of internal organization of the clasts and the matrix” I cannot
understand this sentence. Rewrite.

Modified, as requested.

Line 231 “Similar” not capital

Corrected, thank you.

Line 223-240 I feel that this is a problem. I don’t think that the discussion in this paragraph is not
enough to that the brecciation was caused by seismicity.

As we wrote at line 239-40 of the submitted manuscript, we “suggest” that the brecciation was caused
by seismic activity. The line of evidence that we used to propose this link are stated in the paragraphs
above.

Line 257 “ $0.333 > X_{O_2} > 0.333$ ” I cannot understand the meaning of this inequality.

That was actually correct. Anyhow we changed to \neq for clarity.

Line 241-268 I understand that molar volume of CH₄ is larger than H₂O and CO₂. But the effect of
course depends on CH₄ produced. What is the expected X_{CH_4} ($= CH_4 / (CH_4 + H_2O)$) in fluids?

We report here the response to a similar comment by Reviewer#1:

Raman spectra cannot be used to derive the CH₄/H₂O ratio. However, we used the obtained Raman
spectra to estimate the proportions of CH₄ and H₂ in the gaseous part of the fluid, which is 86% CH₄
and 14% H₂ (no other gaseous molecules detected). Thanks to a software developed in our group
(Boutier et al., submitted), we estimated the conditions for which this proportions are possible within
a COH ternary system at 550 °C and 2 GPa. We found that, at these conditions, the measured CH₄/H₂
ratio is possible for a carbon undersaturated fluid at XO ≈ 0.07, which corresponds to ΔFMQ-7
(previous work estimated ΔFMQ-6 for the CH₄-forming event). At the estimated conditions, the
XH₂O is 0.25, whereas the XCH₄ is 0.65. So, CH₄ was dominant in this aqueous fluid. The carbon-
undersaturated nature of this fluid in a graphite-bearing vein system may be explained by pressure
fluctuations during the breccia formation process, or by disequilibrium conditions, as suggested by
the vein mineralogy. Because these considerations do not take into account possible fluid inclusion
respeciation during exhumation, the revised manuscript now presents some of these results as a
qualitative assessment.

Line 290-291 The fracture networks were sealed by eclogite-facies silicate minerals. This means that
the fluid is aqueous fluids (H₂O-dominated). In this cases, how much CH₄ dissolved in fluids
influenced the hydrofracturing? In addition, what is the source of the silicate-forming aqueous fluids?
If the H₂O is dominant in fluids, the production and accumulation of H₂O fluids also should be
discussed. At the time of fracturing, the ultramafic rocks were completely serpentinized?

We clarified throughout the manuscript that is CH₄-bearing aqueous fluid. The response to a previous
comment should clarify the proportion of CH₄ in the fluid. Regarding the effect of CH₄, our results
show that even a small amount of it has drastic effects in causing overpressure. At the time of
fracturing the rocks were not completely serpentinized, as the Lanzo Massif still displays nowadays
several areas where fresh peridotites are found. Vitale Brovarone et al 2020 Nat comm showed that
the fresh peridotite part of the Lanzo massif shows evidence for high-pressure serpentinization and

reducing conditions. We added this information in the geological setting. Regarding the nature of the
fluid, we implemented the Raman Spectroscopy data presentation and discussion, modifying Fig. 8
and adding a new supplementary figure 15.

Figure 9. Pore fluid factor I wonder whether lowest fluid pressure can be to hydrostatic even at the
2GPa depth.

The reviewer is correct that at such depths pore pressure is likely higher than hydrostatic due to the
relatively low permeability at those depths, even though the entity of such fluctuations at forearc
depths is still not fully understood (see sketch in Fig. 9 of Saffer and Tobin 2011
<https://www.annualreviews.org/doi/10.1146/annurev-earth-040610-133408>).

We modified the figure increasing the lower value of such fluctuations and adding a dashed line, to
express this uncertainty.

**Reviewer 3**

This paper describes samples of a layer of high pressure clinopyroxene (omphacite) rock (given the
rock name of omphacitite) in a host rock of serpentinite (hydrated peridotite). The omphacitite is
highly fractured, consistent with fractures generated by overpressured fluids, and these fractures are
abundantly coated by graphite, thus suggesting that the fluid must have contained methane which was
subsequently reduced to carbon by the presence of hydrogen. The hydrogen is assumed to come from
the serpentinisation process. The aim of the paper is to go beyond this simple description and
demonstrate that such methane-hydrogen-rich fluids are more efficient at generating the seismic
fracturing than hydrous fluids.

The geological context is given as the forearc region of a convergent plate boundary with the
omphacitite described as a metasomatic layer. To provide a better context for the study, it would have
been helpful to have more detail about how this metasomatic layer was formed, even if this was in
the Supplementary information. The references (35-37) are unhelpful in this regard. In Figure 1b the

omphacitite appears to be a vein within serpentinite. What is being metasomatized, or is it all
precipitated from a fluid as implied by the crack-and-seal description of the texture (or does that only
refer to veins within the omphacitite?). Given that the focus in the paper is this omphacitite some
further details of how it was formed would be helpful for the subsequent interpretation of events.

The geological background has been thoroughly revised, following also previous reviewers'
comments. Regarding the cited references, Castelli et al., 1995 (citation n°37 of the submitted
article) suggest that the omphacitite could represent a HP metamorphosed rodingite, stating that
"*rocks of the southern Sesia-Lanzo Zone (metabasites, micaschists and gneisses) are in tectonic*
*contact with the serpentinitized ultramafites of the Lanzo Massif; the contact is decorated by a metre-*
*thick rim of high pressure (omphacite-bearing) to low pressure (diopside-bearing) layered*
*rodingite (Compagnoni et al., 1980).*". However, we found relicts of monazite grains that suggest
that the protolith was at least in part from the continental rocks of the Sesia Zone. Therefore, we
suspect that this omphacite layers formed at the expenses of different protoliths. This is, however,
beyond the scope of this article, as here we do not focus on the formation but on the deformation of
this strong rock type.

The deformation twinning is interpreted as the first 'event'. The discussion of the critical resolved
shear stress (CRSS) for twin formation is referred to jadeite and diopside, commenting that
comparable results were obtained for omphacite. This could be clarified to comment that omphacite
is significantly stronger than jadeite or diopside, whereas the text implies that they have similar CRSS.

The most recent experimental study of Moghadam et al., 2010 suggest comparable results between
jadeite and omphacite, we thus prefer to keep it conservative. In any case, if CRSS for omphacite is
higher, this would imply that the rock underwent higher shear stresses, as the reviewer suggests. Note,
however, that hybrid failure is observed by the microstructures, and this is compatible with
differential stresses in the order of 100 MPa.

There is also discussion in the literature that whether the omphacite is cation ordered (i.e. P2/n) or
disordered (C2/c) makes a difference to the strength. There is no comment about this in the paper,
although diffraction patterns in the TEM could have determined this and perhaps explain why only
the grains near the edges of the larger clasts are twinned. The P,T conditions would suggest that the
omphacite should be ordered. The TEM describes the twins but no antiphase domains (APDs) are
imaged. Perhaps a lost opportunity here.

Although some authors suggested what stated by the reviewer (Dorner & Stockert 2004
<https://doi.org/10.1016/j.tecto.2003.11.008>), untangle what are the implications on strength is still a
controversial matter (see Moghadam et al., 2010) and is beyond the scope of the present article.
Omphacite shows a P2/n component, in agreement with the maximum T condition of metamorphism
estimated for the area (600°C). Unfortunately, to visualize the antiphase domains very specific
imaging conditions are needed that are beyond the scope of this study.

The jadeite + omphacite intergrowth could also be interpreted as a stable co-existence, according to
the Carpenter phase diagram (P2/n + C2/c), rather than as dendritic textures interpreted as fast
precipitation under nonequilibrium conditions. Again the presence of very fine APD's could have
shed some light on this. The interpretation that the intergrowth is dendritic seems to be an important
aspect of the overall interpretation (line 239).

To strengthen our documentation of these structures, we added 2 new figures in the supplementary
(Figures S6 and 7) to document the jadeite and omphacite intergrowths, as also requested by Reviewer
1. Additionally, we also have evidence of cauliflower garnet intergrown with those mineral phases,
in literature interpreted as a result of fast growth under disequilibrium (see reply to previous comment
n°3 of Reviewer 1 and new Figure 4 g,h). Here, we briefly summarize the main points:

- • Rapid growth is suggested also by the presence of garnet with a cauliflower texture intergrown
with jadeite in the matrix (e.g. Altenberger et al. 2013, Clerc et al., 2018, Incel et al. 2020).

*Mancktelow et al. 2022.*). Noteworthy, the cited references found a link between the
cauliflower structure of garnet and fast growth related to seismic activity. Finally, cauliflower
garnet was also related to rapid growth under chemical disequilibrium (*Wilbur, D. E., & Ague,*
*J. J. (2006)*).

Despite these comments, the paper makes a reasonable case that CO₂ bearing fluids may produce
greater overpressures than aqueous solutions (higher wetting angles, larger molar volume) and given
that the evidence is consistent with fluid-induced fracturing, the conclusion that carbon-bearing fluids
could trigger seismic failure is plausible. The paper could be improved with some attention to the
issues raised in this review.

We implemented what requested in the revised version. We built on the constructive comment
provided by all reviewers to strengthen our manuscript. Finally, we do not claim that we found an
unequivocal record of seismic activity occurring at forearc depths, but based on what presented in the
text, on relevant literature, and on what discussed in our replies to reviewers, we find the possible
link between the studied samples and subduction seismicity reasonable and justified..

Minor comment: References 43 and 47 in the list are identical.

Corrected, thank you.

REVIEWERS' COMMENTS

Reviewer #1 (Remarks to the Author):

From my point of view, the authors have carefully revised their manuscript and addressed the comments of the reviewers.

I have just few very minor comments:

- What the authors mean by “fluid overpressure” gets clearer in the revised manuscript. Still, as “fluid overpressure” can refer to a pressure higher than the hydrostatic pressure, i.e., a sublithostatic pressure, but also to “supralithostatic pressures”, or may as well be used for a threshold above which e.g. brecciation occurs and because it is such an important term for the paper, maybe a short explanation of this term with reference to Peacock et al. (2017, [1http://dx.doi.org/10.1016/j.jvolgeores.2017.05.005](http://dx.doi.org/10.1016/j.jvolgeores.2017.05.005)), who addressed the different usage of this term and the importance to clarify it, would be helpful.

- Line 79-80 Please rephrase the sentence, as omphacitite is the rock comprising omphacite, which is the characteristic clinopyroxene of eclogite-facies mafic rocks.

- Line 354: supralithostatic instead of supra-lithostatic

- Line 335: Probably the authors mean “brittle” failure?

Reviewer #2 (Remarks to the Author):

A review of “Methane-rich fluid migration may trigger seismic failure in subduction at forearc depth” by F. Guintoli et al.

I confirmed that the authors have succeeded in improving the manuscripts, and answered all points I suggested, especially on lithological relationship and fluid compositions. I appreciate the efforts of the authors on the revisions.

Minor points:

Line 64 Okamoto et al. (2021) may be useful for showing natural occurrence of fracturing during carbonation in subduction zone.

Okamoto, A., Oyanagi, R., Yoshida, K., Uno, M., Shimizu, H., Satishkumar, M., 2021. Rupture of wet mantle wedge by self-promoting carbonation. *Communications Earth & Environment*, 2, 151.

Line 170 microns => "micron"m

Line 206 and others subscript XCH₄

Line 296-298 cite figures showing the precipitation of graphite.

Line 315 CH₄, H₂O => subscript

Line 356-358 Cite figure 8 here.

**Point-by-point response**

**We used the red colour to reply to the reviewers' comments**

**Reviewer Comments:**

**Reviewer 1**

I have just few very minor comments:

- What the authors mean by “fluid overpressure” gets clearer in the revised manuscript. Still, as “fluid
overpressure” can refer to a pressure higher than the hydrostatic pressure, i.e., a sublithostatic
pressure, but also to “supralithostatic pressures”, or may as well be used for a threshold above which
e.g. brecciation occurs and because it is such an important term for the paper, maybe a short
explanation of this term with reference to Peacock et al. (2017,
<http://dx.doi.org/10.1016/j.jvolgeores.2017.05.005>), who addressed the different usage of this term
and the importance to clarify it, would be helpful.

**We thank the reviewer for this additional comment. The text, at line 287-291, was revised as follows:**
**“The term overpressure is generic(ref) and literature data point to either sublithostatic or**
**supralithostatic conditions. In the present case study, Mohr-Coulomb diagrams (Fig. 7) clearly point**
**to supralithostatic fluid pressure. In order to assess the potential of CH₄-bearing aqueous fluids for**
**supralithostatic hydrofracturing at subduction zone conditions, we performed...”. Elsewhere in the**
**text, we used either the generic term overpressure or specific terminology upon context (e.g., generic**
**terms for the abstract).**

- Line 79-80 Please rephrase the sentence, as omphacitite is the rock comprising omphacite, which is
the characteristic clinopyroxene of eclogite-facies mafic rocks.

We rephrased in “In this contribution, we show evidence of brecciation of omphacitite, a rock rich in
omphacite that represent the characteristic clinopyroxene of eclogite-facies mafic rocks.”.

- Line 354: supralithostatic instead of supra-lithostatic

Corrected.

- Line 335: Probably the authors mean “brittle” failure?

We added brittle as suggested.

**Reviewer 2**

A review of “Methane-rich fluid migration may trigger seismic failure in subduction at forearc depth”
by F. Guintoli et al.

I confirmed that the authors have succeeded in improving the manuscripts, and answered all points I
suggested, especially on lithological relationship and fluid compositions. I appreciate the efforts of
the authors on the revisions.

Minor points:

Line 64 Okamoto et al. (2021) may be useful for showing natural occurrence of fracturing during
carbonation in subduction zone. Okamoto, A., Oyanagi, R., Yoshida, K., Uno, M., Shimizu, H.,
Satishkumar, M., 2021. Rupture of wet mantle wedge by self-promoting carbonation.
Communications Earth & Environment, 2, 151.

We added the suggested reference, thank you.

Line 170 microns => "micron"m

Written microns for consistency.

Line 206 and others subscript X_{CH_4}

Corrected.

Line 296-298 cite figures showing the precipitation of graphite.

Added citation to Fig. 1e, as requested.

Line 315 CH₄, H₂O => subscript

Corrected, thank you.

Line 356-358 Cite figure 8 here.

Done, thank you.
